# Mapping the transcriptional diversity of genetically and anatomically defined cell populations in the mouse brain

Ken Sugino[1]*, Erin Clark[2†], Anton Schulmann[1†], Yasuyuki Shima[2], Lihua Wang[1], David L Hunt[1], Bryan M Hooks[1], Dimitri Tränkner[1], Jayaram Chandrashekar[1], Serge Picard[1], Andrew L Lemire[1], Nelson Spruston[1], Adam W Hantman[1]*, Sacha B Nelson[2]*

[1]Janelia Research Campus, Ashburn, United States; [2]Brandeis University, Waltham, United States

**Abstract** Understanding the principles governing neuronal diversity is a fundamental goal for neuroscience. Here, we provide an anatomical and transcriptomic database of nearly 200 genetically identified cell populations. By separately analyzing the robustness and pattern of expression differences across these cell populations, we identify two gene classes contributing distinctly to neuronal diversity. Short homeobox transcription factors distinguish neuronal populations combinatorially, and exhibit extremely low transcriptional noise, enabling highly robust expression differences. Long neuronal effector genes, such as channels and cell adhesion molecules, contribute disproportionately to neuronal diversity, based on their patterns rather than robustness of expression differences. By linking transcriptional identity to genetic strains and anatomical atlases, we provide an extensive resource for further investigation of mouse neuronal cell types.
DOI: https://doi.org/10.7554/eLife.38619.001

*For correspondence:
ken.sugino@gmail.com (KS);
hantmana@janelia.hhmi.org
(AWH);
nelson@brandeis.edu (SBN)

†These authors contributed
equally to this work

Competing interests: The
authors declare that no
competing interests exist.

Reviewing editor: Chris P
Ponting, University of Edinburgh,
United Kingdom

## Introduction

The extraordinary diversity of vertebrate neurons has been appreciated since the proposal of the neuron doctrine (*Ramon y Cajal, 1894*). Classically, this diversity was characterized by neuronal morphology, physiology, and circuit connectivity, but increasingly, defined genetically through driver and reporter strains (*Gong et al., 2003*; *Madisen et al., 2010*; *Taniguchi et al., 2011*; *Shima et al., 2016*) or genomically by their genome-wide expression profiles. The first genome-wide studies of mammalian neuronal diversity employed in situ hybridization or microarrays (*Sugino et al., 2006*; *Doyle et al., 2008*), while more recent studies have utilized advances in single-cell (SC) RNA-seq (*Zeisel et al., 2015*; *Zeisel et al., 2018*; *Tasic et al., 2016*; *Tasic et al., 2018*; *Paul et al., 2017*). In theory, SC RNA-seq can be applied in an unbiased fashion to discover all cell types that comprise a tissue, but manipulation of these cell types to better understand their biological composition and function often require the use of genetic tools such as mouse driver strains. Differences in techniques for cell isolation, library preparation or clustering of single cell profiles have not yet led to a consensus view of the number or identity of the neuronal cell types comprising most parts of the mouse nervous system. Furthermore, the relationship between cell populations defined transcriptionally and those that can be specified genetically and anatomically using existing strains has received far less attention (though see *Tasic et al., 2018*).

Here, we attempt to strengthen the link between genomically and genetically defined cell types in the mouse brain by performing RNA-seq on a large set of genetically identified and fluorescently labeled neurons from micro-dissected brain regions. In total, we profiled 179 sorted neuronal

populations and 15 nonneuronal populations. Because each sample of sorted cells may contain more than one 'atomic' cell type, we refer to these as genetically- and anatomically-identified cell populations (GACPs). To assess homogeneity, we quantitatively compared our sorted cell populations to publicly available single-cell datasets, which revealed a comparable level of homogeneity, but a much lower level of noise in the sorted population profiles.

Although neuronal diversity has long been recognized, the question of how this diversity arises has not been addressed sufficiently in a genomic context (*Arendt et al., 2016*; *Muotri and Gage, 2006*). We identify two different sets of genes that distinguish GACPs based on the robustness or pattern of their expression differences. The most robust expression differences are those of homeobox transcription factors. These genes also have the lowest transcriptional noise suggesting differential chromatin regulation. Chromatin accessibility measurements reveal that the promoters and gene bodies of these genes are indeed more closed. In contrast, the genes capable of distinguishing the largest numbers of GACPs are neuronal effector genes like receptors, ion channels and cell adhesion molecules. Interestingly, genes defined by the robustness and patterns of their expression differences also differ in their transcript length. Genes with robust, low-noise expression tend to be shorter, while genes with the greatest capacity to distinguish populations tend to be longer.

Here, we provide important new resources for mapping brain cell types including a large set of low-noise profiles from genetically identified neurons, anatomical maps of their distributions, and a method to compare and contextualize single-cell RNA-seq datasets. We implement a novel strategy to mine information from large surveys of cell types, and demonstrate the utility of this strategy in generating specific biological insights into the genes contributing to neuronal diversity.

## Results

### A dataset of genetically identified neuronal transcriptomes

To identify genes contributing most to mammalian neuronal diversity, we collected transcriptomes from 179 genetically and anatomically identified populations of neurons and 15 populations of non-neuronal cells in mice (*Table 1*; *Figure 1*; *Figure 1—figure supplement 1*; *Supplementary file 1*,*2*). The great majority (186/194) were identified both genetically and anatomically, with the remaining identified only anatomically, by their location and projection patterns. Each collected population represents a group of fluorescently labeled cells dissociated and sorted from a specific micro-dissected region of the mouse brain or other tissue. The pipeline for collecting GACP transcriptomes is depicted in *Figure 1A* (see Materials and methods for additional details). Mouse lines were first characterized by generating a high-resolution atlas of reporter expression (*Figure 1B*) then, regions containing labeled cells with uniform morphology were chosen for sorting and RNA-seq. In total, we sequenced 2.3 trillion bp in 565 libraries. This effort (NeuroSeq) constitutes the largest and most diverse single collection of genetically identified cell populations profiled by RNA-seq. The raw data is deposited to NCBI GEO (GSE79238). The processed data, including anatomical atlases, RNA-seq coverage, and TPM are available at http://neuroseq.janelia.org (*Figure 1C*).

To determine the sensitivity of our transcriptional profiling, we used ERCC spike-ins. Amplified RNA libraries had an average sensitivity (50% detection) of 23 copy*kbp of ERCC spike-ins across all libraries (*Figure 1D*). Since manually sorted samples had $132 \pm 16$ cells (mean $\pm$ SEM), this indicates our pipeline had the sensitivity to detect a single copy of a transcript per cell 80% of the time. This high sensitivity allowed for deep transcriptional profiling in our diverse set of cell populations.

To assess the extent of contamination in the dataset, we checked expression levels of marker genes for several nonneuronal cell populations (*Figure 1—figure supplement 2B*). As previously shown (*Okaty et al., 2011*), manual sorting produced, in general, extremely clean data.

To assess the homogeneity of the sorted, pooled samples, we compared our datasets to publicly available single cell (SC) datasets. To compare across different datasets, we used a method based on linear decomposition by non-negative least squares (NNLS) (See *Figure 2* and *Figure 2—figure supplement 1–6*). This method tests the degree to which individual profiles can be decomposed into linear mixtures of profiles from another dataset. Such mixtures or impurities can arise in at least two ways (*Figure 2A*): by pooling similar cell types prior to sequencing in the case of sorted datasets, or by pooling similar profiles after sequencing, at the clustering stage, in the case of SC datasets. Although NNLS is a widely used decomposition procedure, it has not previously been applied

**Table 1.** Summary of profiled samples.

| | Region/type | Transmitter | #groups | Subregions | #samples |
|---|---|---|---|---|---|
| CNS neurons | Olfactory (OLF) | glu | 10 | AOBmi, MOBgl, PIR, AOB, COAp | 30 |
| | | GABA | 4 | AOBgr, MOBgr, MOBmi | 11 |
| | Isocortex | glu | 22 | VISp, AI, MOp5, MO, VISp6a, SSp, SSs, ECT, ORBm, RSPv | 68 |
| | | GABA | 3 | Isocortex, SSp (Sst+, Pvalb+) | 7 |
| | | glu,GABA | 1 | RSPv | 3 |
| | Subplate (CTXsp) | glu | 1 | CLA | 4 |
| | Hippocampus (HPF) | glu | 24 | CA1, CA1sp, CA2, CA3, CA3sp, DG, DG-sg, SUBd-sp, IG | 65 |
| | | GABA | 4 | CA3, CA, CA1 (Sst+, Pvalb+) | 12 |
| | Striatum (STR) | GABA | 12 | ACB, OT, CEAm, CEAl, islm, isl, CP | 33 |
| | Pallidum (PAL) | GABA | 1 | BST | 4 |
| | Thalamus (TH) | glu | 11 | PVT, CL, AMd, LGd, PCN, AV, VPM, AD | 29 |
| | Hypothalamus (HY) | glu | 11 | LHA, MM, PVHd, SO, DMHp, PVH, PVHp | 36 |
| | | GABA | 4 | ARH, MPN, SCH | 15 |
| | | glu,GABA | 2 | SFO | 3 |
| | Midbrain (MB) | DA | 2 | SNc, VTA | 5 |
| | | glu | 2 | SCm, IC | 6 |
| | | 5HT | 2 | DR | 10 |
| | | GABA | 1 | PAG | 4 |
| | | glu,DA | 1 | VTA | 3 |
| | Pons (P) | glu | 7 | PBl, PG | 22 |
| | | NE | 1 | LC | 2 |
| | | 5HT | 2 | CSm | 7 |
| | Medulla (MY) | GABA | 7 | AP, NTS, MV, NTSge, DCO | 18 |
| | | glu | 6 | NTSm, IO, ECU, LRNm | 20 |
| | | ACh | 2 | DMX, VII | 6 |
| | | 5HT | 1 | RPA | 3 |
| | | GABA,5HT | 1 | RPA | 4 |
| | | glu,GABA | 1 | PRP | 3 |
| | Cerebellum (CB) | GABA | 10 | CUL4, 5mo, CUL4, 5pu, CUL4, 5gr, PYRpu | 25 |
| | | glu | 4 | CUL4, 5gr, NODgr | 10 |
| | Retina | glu | 5 | ganglion cells (MTN, LGN, SC projecting) | 14 |
| | Spinal Cord | glu | 1 | Lumbar (L1-L5) dorsal part | 3 |
| | | GABA | 4 | Lumbar (L1-L5) dorsal part, central part | 12 |
| PNS | Jugular | glu | 2 | (TrpV1+) | 7 |
| | Dorsal root ganglion (DRG) | glu | 2 | (TrpV1+, Pvalb+) | 5 |
| | Olfactory sensory neurons (OE) | glu | 4 | MOE,VNO | 9 |
| nonneuron | Microglia | | 2 | MOp5(Isocortex),UVU(CB) (Cx3cr1+) | 6 |
| | Astrocytes | | 1 | Isocortex (GFAP+) | 4 |
| | Ependyma | | 1 | Choroid Plexus | 2 |
| | Ependyma | | 2 | Lateral ventricle (Rarres2+) | 6 |
| | Epithelial | | 1 | Blood vessel (Isocortex) (Apod+, Bgn+) | 3 |
| | Epithelial | | 1 | olfactory epithelium | 2 |
| | Progenitor | | 1 | DG (POMC+) | 3 |
| | Pituitary | | 1 | (POMC+) | 3 |

*Table 1 continued on next page*

*Table 1 continued*

| | Region/type | Transmitter | #groups | Subregions | #samples |
|---|---|---|---|---|---|
| non brain | Pancreas | | 2 | Acinar cell, beta cell | 7 |
| | Myofiber | | 2 | Extensor digitorum longus muscle | 7 |
| | Brown adipose cell | | 1 | Brown adipose cell from neck. | 4 |
| | | total | 194 | | 565 |

DOI: https://doi.org/10.7554/eLife.38619.006

to expression profiles. Therefore, we performed a number of control experiments to validate its use. First, we cross-validated the decompositions by dividing each dataset in half and testing the ability to decompose one half by the other (*Figure 2—figure supplement 1*). This revealed that some NeuroSeq samples had overlapping coefficients and so could not be well distinguished. For example, pairs of populations identified in layer 2/3 of two different regions in the same strain (AI.L23_glu_P157/ORBm.L23_glu_P157) or by retrogradely labeled cells in the same layer and region from two different targets (SSp.L23_glu_M1.inj/SSp.L23_glu_S2.inj and SSp.L5_glu_BPn.inj/SSp.L5_glu_IRT.in) were hard to distinguish. On the other hand, overlapping coefficients were also present for some pairs of cell populations in the SC datasets (such as Oligo Serpinb1a/Oligo Synpr in the Tasic dataset and MGL1/MGL2/MGL3 in the Zeisel dataset). On average the purity, defined as how well a single sample can be decomposed into the most closely corresponding sample, was similar across the three datasets (*Figure 2—figure supplement 1D*). As a second control, we demonstrated that NNLS decomposition could be used to recover the numbers of cell types isolated from distinct strains in a SC dataset, after mixing these profiles together, despite the fact that this information was not included in the fitting procedure (*Figure 2—figure supplement 2*). Finally, NNLS (*Figure 2B,C*) produced comparable or cleaner decompositions than a competing Random Forest algorithm (*Figure 2—figure supplement 6*). These results indicate that NNLS can be used to reliably decompose mixtures of cellular profiles. Similar average coefficients (i.e. similar purity) were obtained for decompositions of the NeuroSeq data by SC datasets and by decomposing these datasets by each other (*Figure 2*, *Figure 2—figure supplement 3–6*). Hence our decomposition results indicate that although heterogeneity may exist in some of our sorted samples, it is comparable to the inaccuracies introduced by clustering SC profiles.

Since merging or splitting of closely related clusters either prior to sequencing or during the clustering process can lead to poor discrimination between samples, we also measured the separability of cell population profiles obtained in each study (*Figure 2—figure supplement 7*). As expected, the clusters of sorted population samples, which are averages across one hundred cells or more, were much more cleanly separable than SC clusters. Taken together, NNLS decomposition and separability provide a quantitative framework for assessing the trade-offs between homogeneity and reproducibility when measuring population transcriptomes from GACPs and SCs.

To demonstrate the utility of the dataset, made possible by its broad sampling of neuronal populations, we extracted pan-neuronal genes (genes expressed commonly in all neuronal populations but expressed at lower levels or not at all in nonneuronal cell populations; *Figure 1—figure supplement 3*). Here, broad sampling of cell populations is essential to avoid false positives (*Zhang et al., 2014b*; *Mo et al., 2015*; *Stefanakis et al., 2015*). Because of the high sensitivity and low noise, we were able to be conservative and exclude genes expressed in most but not all neuron types. Extracted pan-neuronal genes contain well-known genes such as *Eno2* (Enolase2), which is the neuronal form of Enolase required for the Krebs cycle, *Slc2a3* (chloride transporter) required for inhibitory transmission, and *Atp1a3* (ATPase Na+/K + transporting subunit alpha 3), which belongs to the complex responsible for maintaining electrochemical gradients across the membrane, as well as genes not previously known to be pan-neuronal, such as *2900011O08Rik* (now called Migration Inhibitory Protein; *Zhang et al., 2014a*). Synaptic genes are often differentially expressed among neurons, but interestingly, some were included in this pan-neuronal list such as *Syn1, Stx1b, Stxbp1, Sv2a*, and *Vamp2*. These appear to be common synaptic components, and highlight essential parts of these complexes. Thus, the dataset should be useful for many other applications, especially those requiring comparisons across a wide variety of neuronal cell types.

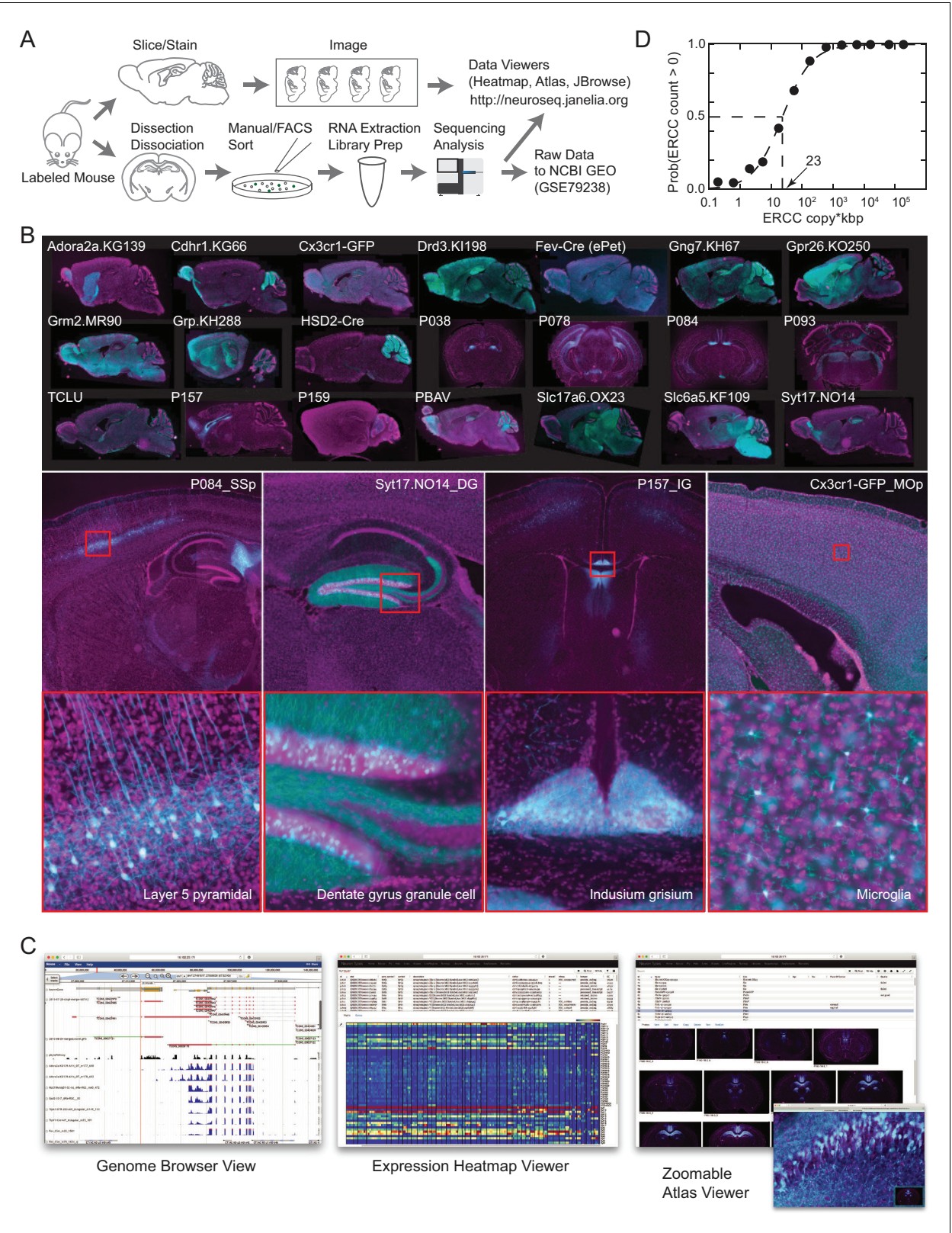

**Figure 1.** The NeuroSeq dataset. (**A**) Schema of pipeline for anatomical and genomic data collection. (**B**) Example sections from atlases at low (top), medium (middle) and high (bottom) magnifications. (**C**) Web tools available at http://neuroseq.janelia.org (**D**) Sensitivity of library preparation measured from ERCC detection across all libraries. The 50% detection sensitivity of the assay itself was 23 copy*kbp.

DOI: https://doi.org/10.7554/eLife.38619.002

*Figure 1 continued on next page*

*Figure 1 continued*

The following figure supplements are available for figure 1:

**Figure supplement 1.** GACP samples.
DOI: https://doi.org/10.7554/eLife.38619.003
**Figure supplement 2.** Quality control measures.
DOI: https://doi.org/10.7554/eLife.38619.004
**Figure supplement 3.** Pan-neuronal genes.
DOI: https://doi.org/10.7554/eLife.38619.005

## Metrics to quantify diversity

Analysis of expression differences between individual groups is the basis of most profiling efforts. Variance-based metrics, such as Analysis of Variance (ANOVA) F-Value, or coefficient of variation (CV) are commonly used for this purpose. However, these metrics are jointly affected by the pattern of differential expression and the robustness of the differences, and so cannot readily separate these two features (*Figures 3* and *4*; *Figure 3—figure supplement 1*). Since these features may differ in their biological significance, we searched for the simplest way to quantitatively separate them. This led us to adopt two easily calculated variants of widely used metrics for differential expression and fold-change.

To quantify the contribution of each gene to cell type diversity, we measured the fraction of cell population pairs in which the gene is differentially expressed. (For differential analysis, the limma-voom framework was used, see Materials and methods). This differentially expressed fraction (DEF) is closely related to the Gini-Simpson diversity index (*Simpson, 1949*) widely used in ecology to measure species diversity in a community (see Appendix 1). DEF ranges from 0 to 1. The maximum observed value of 0.65 indicates that the gene distinguishes 65% of the pairs, while a value of 0 indicates that the gene distinguishes none (i.e. it is expressed at similar levels in all cells). DEF is easy to calculate and approximates the mutual information (MI) between expression levels and cell populations (Appendix 1).

The robustness of an expression difference depends on its magnitude relative to the underlying noise. Robustness is often quantified as a Signal-to-Noise Ratio (SNR). Since the signals we are interested in are the gene expression differences distinguishing cell types, we computed the ratio of the mean fold-change expression differences between distinguished pairs to the mean fold-change between undistinguished pairs. This fold-change ratio (FCR) indicates the robustness of pair distinctions but is independent of the number of pairs distinguished. High FCR genes robustly distinguish cell populations and are therefore suitable as 'marker genes'.

Unlike DEF and FCR, variance-based methods like ANOVA F-values and CV are either affected by both MI and SNR (ANOVA; *Figure 4A–C* and *Figure 3—figure supplement 1*) or by neither (CV; *Figure 3—figure supplement 1*). The fact that ANOVA does not distinguish between information content and SNR can be appreciated from the fact that the most significant ANOVA genes (*Figure 4A–C*) include both high DEF and high FCR genes. Therefore, DEF and FCR are useful because they provide independent measures of the robustness and magnitude of differential expression between cell populations.

To determine the types of genes most differentially expressed (highest DEF) and most robustly different (highest FCR) between cell populations, we performed over-representation analysis using the HUGO Gene Groups (*Braschi et al., 2018*, *Figure 4D,E*). The most robust expression differences (highest FCR) were those of homeobox transcription factors (TFs) and G-protein coupled receptors (GPCRs; *Figure 4D*). High DEF genes are enriched for neuronal effector genes including receptors, ion channels and cell adhesion molecules (*Figure 4E*). High FCR and High DEF enrichments were based on the HUGO gene groups, but similar results were obtained using the PANTHER gene families (*Mi et al., 2017*) and Gene Ontology annotations (*Ashburner et al., 2000*, *Figure 4—figure supplement 1*). In the case of the high FCR genes, the Gene Ontology categories differed, since this ontology lacks a separate category for homeobox transcription factors. Instead multiple parent categories (e.g. sequence-specific DNA binding, RNA polymerase II regulatory region DNA binding etc.) were overrepresented.

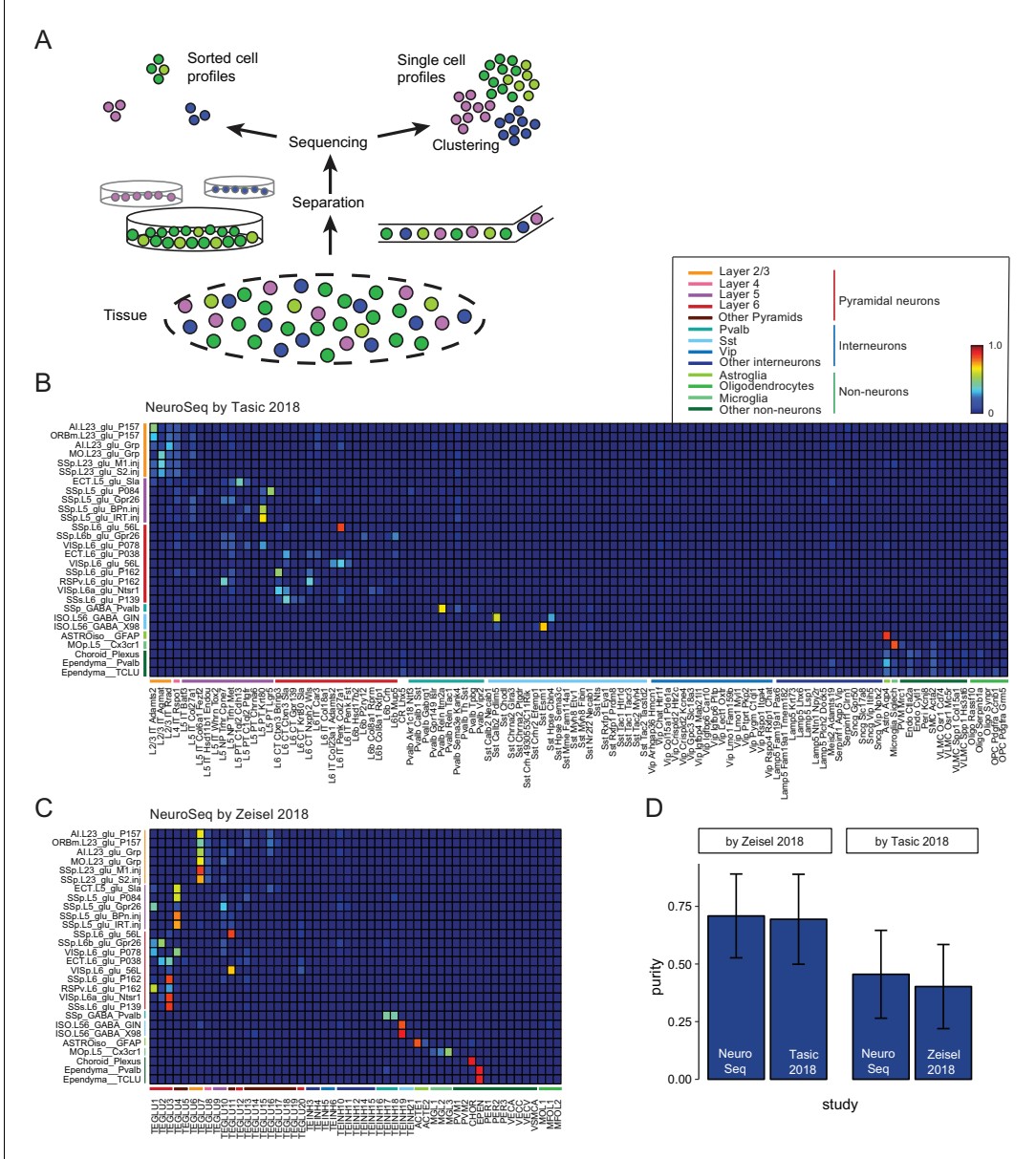

**Figure 2.** Decomposition by non-negative least squares (NNLS) fitting. (**A**) Diagram illustrating potential sources of heterogeneity at the separation phase in profiles from sorted cells (left) or at the clustering phase in profiles from single cells (right). (**B,C**) NNLS coefficients of NeuroSeq cell populations decomposed by two scRNA-seq datasets: (*Tasic et al., 2018*; *Zeisel et al., 2018*). (**D**) Mean purity scores for NeuroSeq and SC datasets. The purity score for a sample is defined as the ratio of the highest coefficient to the sum of all coefficients. Error bars are Std. Dev.

DOI: https://doi.org/10.7554/eLife.38619.007

The following figure supplements are available for figure 2:

**Figure supplement 1.** Self decompositions by NNLS.

DOI: https://doi.org/10.7554/eLife.38619.008

**Figure supplement 2.** A validation of NNLS decomposition.

DOI: https://doi.org/10.7554/eLife.38619.009

**Figure supplement 3.** NNLS decomposition of SC datasets: Tasic by Zeisel.

DOI: https://doi.org/10.7554/eLife.38619.010

**Figure supplement 4.** NNLS decomposition of SC datasets: Zeisel by Tasic.

DOI: https://doi.org/10.7554/eLife.38619.011

**Figure supplement 5.** NNLS decomposition of interneuron datasets.

DOI: https://doi.org/10.7554/eLife.38619.012

*Figure 2 continued on next page*

**Figure supplement 6.** Random forest decomposition.
DOI: https://doi.org/10.7554/eLife.38619.013
**Figure supplement 7.** Separability of cell population clusters.
DOI: https://doi.org/10.7554/eLife.38619.014

Thus, using these two simple metrics we identify synaptic and signaling genes as the most differentially expressed, and homeobox TFs and GPCRs as the most robustly distinguishing families of genes. These two categories of genes drive neuronal diversity by endowing neuronal cell types with specialized signaling and connectivity phenotypes, and by orchestrating cell-type-specific patterns of transcription. In addition, their distinct contributions to distinguishing neuronal types suggests possible differences in the regulation of these two categories of genes.

## Homeobox TFs have the highest SNRs and can form a combinatorial code for cell populations

FCR, like SNR, is a ratio between signal and noise, and so can reflect high expression levels in most ON cell types (high signal), low expression levels in most OFF cell types (low noise), or both. Homeobox genes are not among the most abundantly expressed genes. Their average expression levels (~30 FPKM) are significantly lower than, for example, those of neuropeptides (~90 FPKM). This suggests that the high FCRs of homeobox TFs depend more on low noise than high signal. In fact, many

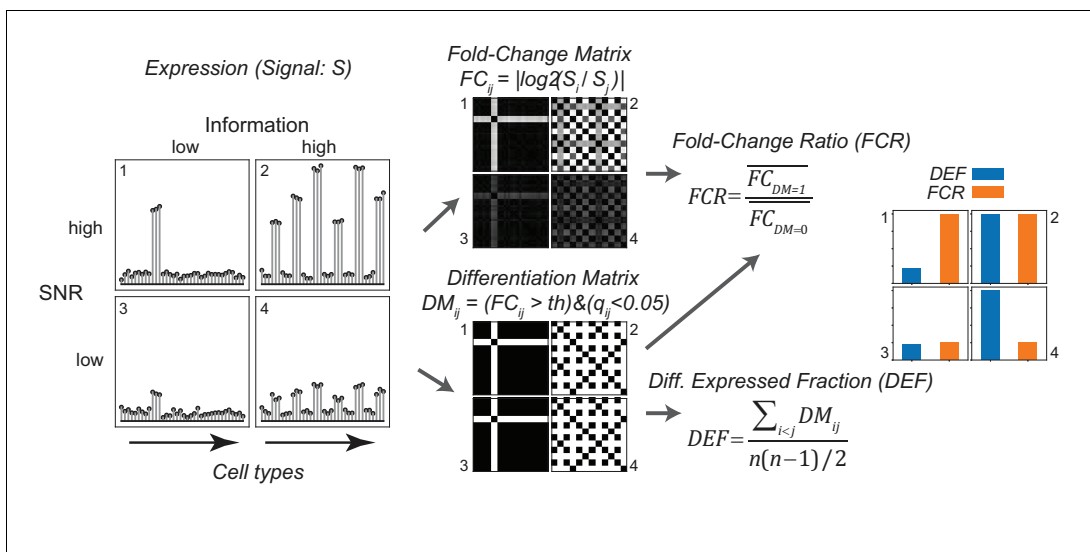

**Figure 3.** Gene expression metrics related to information content and robustness (Left) Cartoon illustrating the process of calculating fold-change ratio (FCR) and differentially expressed fraction (DEF) for four different hypothetical genes that differ in the information content (2 and 4 vs. 1 and 3) and signal-to-noise ratio (SNR; 1 and 2 vs. 3 and 4) of their expression patterns across cell populations. (Middle) Expression signals are used to construct matrices for each gene of the log fold-changes between populations (fold-change matrix) and the distinctions between populations based on those differences (Differentiation Matrix; DM; see Materials and methods). (Right) The differentially expressed fraction (DEF) is the fraction of the total pairs of cell populations distinguished (i.e. of nonzero values in DM excluding diagonal). The fold-change ratio (FCR) is the average expression difference between distinguished pairs divided by the average expression difference between undistinguished pairs. Orange and blue bars show that the resulting DEF and FCR calculations capture the variations in information and SNR across the four genes.
DOI: https://doi.org/10.7554/eLife.38619.015

The following figure supplement is available for figure 3:

**Figure supplement 1.** Simulated data reveal features of expression metrics.
DOI: https://doi.org/10.7554/eLife.38619.016

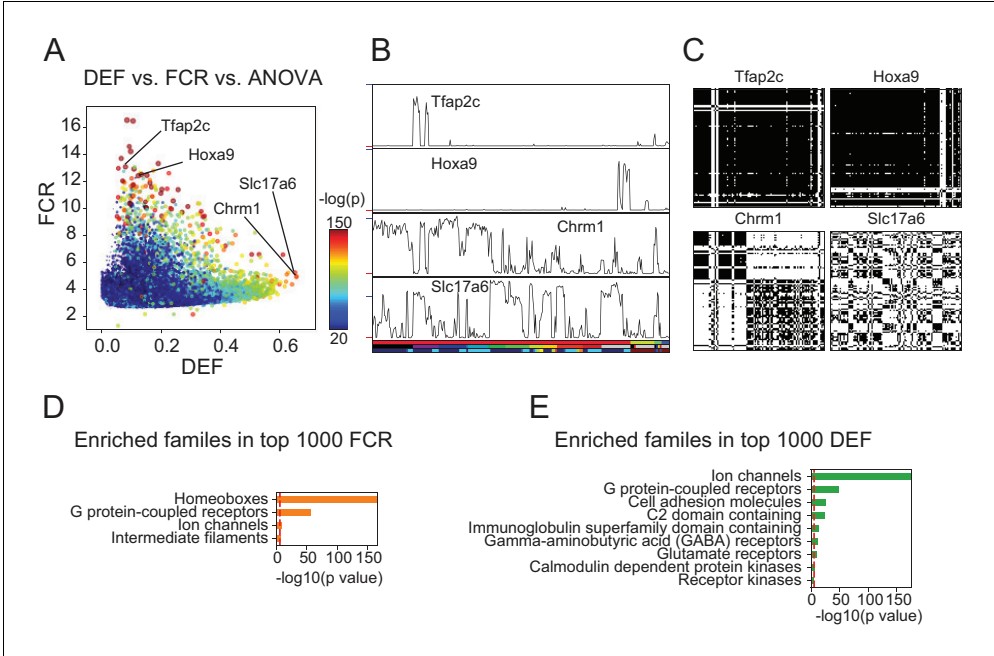

**Figure 4.** DEF and FCR capture distinct aspects of expression diversity related to information content and robustness. (**A**) Highly variable genes (warm colored dots; color scale shows significance of ANOVA across cell populations) include both genes with high FCR and low DEF (like Tfap2c and Hoxa9) and genes with lower FCR and high DEF (like Chrm1 and Slc17a6). (**B**) Expression profiles of example genes labeled in A. Sample key in horizontal color bar as in *Figure 1—figure supplement 1–3*. Red ticks at left indicate 0; Vertical scale is $log_2(FPKM + 1)$; blue ticks = 6) (**C**) DMs for example genes, calculated as shown in *Figure 3*. (**D**, **E**) HUGO gene groups enriched in the top 1000 FCR and top 1000 DEF genes. Red lines indicate the p = $10^{-5}$ threshold used to judge significance.

DOI: https://doi.org/10.7554/eLife.38619.017

The following figure supplement is available for figure 4:

**Figure supplement 1.** PANTHER and GO enrichment analysis for high FCR and high DEF genes.

DOI: https://doi.org/10.7554/eLife.38619.018

---

homeobox TFs have uniformly low expression in OFF cell types (*Figure 5A* top). We quantified this 'OFF noise' for all genes and found that homeobox genes are enriched among genes that have both low OFF noise and at least moderate ON expression levels (red dashed region in *Figure 5B*; see also *Figure 5—figure supplements 1* and *2*). Homeobox genes were not enriched in a group of high OFF noise genes (blue dashed region in *Figure 5B*; data not shown) that was matched for maximum expression level (*Figure 5—figure supplement 1C*). The enrichment of homeoboxes was also observable in two of the single-cell datasets encompassing multiple brain regions (*Figure 5—figure supplement 3*).

Tight control of expression may reflect closed chromatin. To test this, we measured chromatin accessibility using ATAC-seq (see Materials and methods). As expected, compared to high-noise genes (*Figure 5C* bottom), genes with low OFF noise had fewer and smaller peaks within the vicinity of their transcription start site (TSS) and gene body (*Figure 5C* top, *Figure 5D*), consistent with the idea that chromatin accessibility contributes to their low OFF noise. Functionally, the tight control of homeobox TF expression levels may reflect their known importance as determinants of cell identity, and that establishing and maintaining robust differences between cell types may require tight ON/ OFF regulation rather than graded regulation.

Homeobox containing TFs can be subdivided into subfamilies based on their structure. The different homeobox subfamilies differed in their OFF noise and hence in their FCR values. Some families (e.g. HOXL, NKL, PRD) had very low OFF noise and high FCR, while others (e.g. CERS, PROS, CUT) had higher OFF noise and lower FCR (*Figure 5—figure supplement 4*).

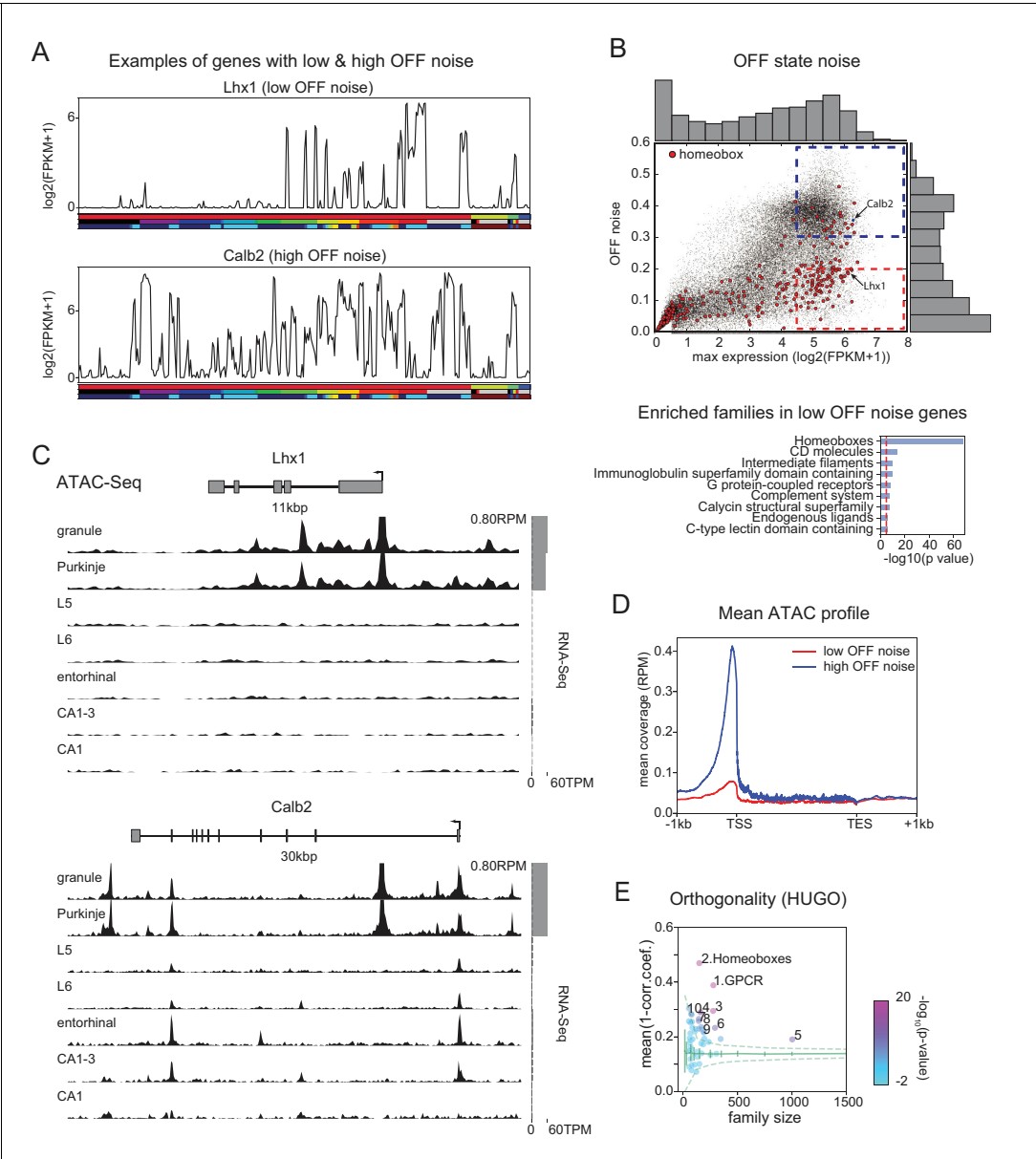

**Figure 5.** Mechanisms contributing to low noise and high information content of homeobox TFs. (**A**) Example expression patterns of a LIM class homeobox TF (Lhx1) and a calcium binding protein (Calb2) with similar overall expression levels. Sample key as in *Figure 1—figure supplement 1– 3*. (**B**) (upper) OFF state noise (defined as standard deviation (std) of samples with FPKM<1) plotted against maximum expression. (lower) HUGO gene groups enriched in the region indicated by red dashed box in the upper panel (see *Figure 5—figure supplement 1* for PANTHER and Gene Ontology enrichments). (**C**) Average (replicate n = 2) ATAC-seq profiles for the genes shown in A. Some peaks are truncated. Expression levels are plotted at right (gray bars). (**D**) Length-normalized ATAC profile for genes with high (> 0.3, blue dashed box in B, n = 853) and low (< 0.2, red dashed box in B, n = 1643) OFF state expression noise. (**E**) Each circle represents the orthogonality of expression patterns calculated using HUGO gene groups. Orthogonality is a measure of the degree of non-redundancy in a set of expression patterns. Since the dispersion of orthogonality depends on family size, results are compared to orthogonality calculated from randomly sampled groups of genes (green solid lines: mean and Std. Dev.; green dashed lines: 99% confidence interval). Families, Z-scores, family size: 1. GPCR: 17.1, n = 277; 2. Homeoboxes: 16.6, n = 148; 3. Ion channels: 10.7, n = 275; 4. C2 domain containing: 7.8, n = 159; 5. Zinc fingers: 6.9, n = 1002; 6. Immunoglobulin superfamily domain containing: 6.7, n = 292; 7. PDZ domain containing: 6.3, n = 144; 8. Fibronectin type III domain containing: 5.9, n = 143; 9. Endogenous ligands: 5.1, n = 165; 10. Basic helix-loop-helix proteins: 4.9, n = 77.

DOI: https://doi.org/10.7554/eLife.38619.019

The following figure supplements are available for figure 5:

**Figure supplement 1.** Properties of Low OFF noise genes.
DOI: https://doi.org/10.7554/eLife.38619.020

*Figure 5 continued on next page*

*Figure 5 continued*

**Figure supplement 2.** Homeobox TFs form a combinatorial code.
DOI: https://doi.org/10.7554/eLife.38619.021
**Figure supplement 3.** OFF noise in single-cell datasets.
DOI: https://doi.org/10.7554/eLife.38619.022
**Figure supplement 4.** OFF noise and gene length in Homeobox subfamilies.
DOI: https://doi.org/10.7554/eLife.38619.023

The ability of gene families to provide information about cell identities reflects both how informative individual family members are, and the relationships between them. If the information across family members is independent, the overall information is increased relative to the case in which multiple members contain redundant information. This aspect of 'family-wise' information is not captured by 'gene-wise' metrics like mean DEF, or by enrichment analysis (*Figure 4D,E*). One means of capturing the additive, non-redundancy within a gene family is to measure the orthogonality of expression patterns among the member genes. This analysis (*Figure 5E*) reveals that homeobox TFs and GPCRs have the greatest orthogonality between cell types among HUGO groups (as well as in PANTHER families, *Figure 5—figure supplement 1E*). Related to this, we found that the homeobox family can distinguish more than 99% of GACP pairs, suggesting these TFs comprise a combinatorial code for the cell populations profiled. To illustrate this, we computed the minimum set of homeobox TFs needed to distinguish the populations studied and found that a set of as few as 8 could distinguish 99% of GACP pairs (*Figure 5—figure supplement 2B*). Combinatorial codes could also be produced from other highly orthogonal gene families, as illustrated for GPCRs *Figure 5—figure supplement 2C*). As illustrated in these heat maps, expression differences for homeobox TFs had higher contrast, consistent with the fact that individually, homeobox TFs have the highest FCR (*Figure 4D*) and lowest OFF noise (*Figure 5B*). In summary, we found that many homeobox genes are expressed with a very high SNR and are one of the groups of genes with the most orthogonal expression patterns. This suggests that, similar to other tissues (*Pereira et al., 2015*; *Gendrel et al., 2016*; *Zheng et al., 2015*; *Dasen and Jessell, 2009*; *Philippidou and Dasen, 2013*), homeobox TFs play an important role in specifying cell types in the brain.

## Diversity arising from alternative splicing

Alternative splicing is known to increase transcriptome diversity (*Andreadis et al., 1987*). To assess the contribution of alternative splicing to diversifying transcriptomes across cell populations, we quantified the branch probabilities at each alternative splice donor site within each gene (*Figure 6A* top). The branch probabilities at each donor site are the relative frequencies with which particular splice acceptors are chosen, and can be estimated from observed junction read counts. Branch probabilities are highly bimodal (*Figure 6A* bottom), suggesting that most branch point choices are made consistently, in an all-or-none fashion, for any given cell population.

To test the significance of differential splicing across cell populations, we utilized a statistical test based on the Dirichlet-Multinomial distribution and the log-likelihood ratio test, developed in Leaf-Cutter (*Li et al., 2018*). We used pair-wise differential expression of each branch to calculate a branch DEF, much as we previously calculated the differentially expressed fraction (DEF) from expression values (*Figure 3*). Examples of branches with high DEFs are shown in *Figure 6B*. The list includes known examples like the site of the flip and flop variants of the AMPA receptor subunit *Gria2* (*Sommer et al., 1990*). Another previously known example is the splicing regulator muscleblind like splicing factor 2 (*Mbnl2*), which is known to regulate splicing in the developing brain (*Charizanis et al., 2012*) and is known to be spliced at multiple sites, including the one shown in *Figure 6B* (*Pascual et al., 2006*).

In order to determine which families of genes are highly differentially spliced, we computed a splice DEF per gene by combining the ability of a gene's alternatively spliced sites to distinguish a pair of samples (i.e. a pair is distinguished by a gene if any alternatively spliced site in the gene can distinguish the pair). Using this combined splice DEF, we found that RNA binding proteins, especially splicing related factors (such as *Pcbp2* and *Mbnl2*) are highly alternatively spliced among neuronal cell types (*Zheng and Black, 2013*), but over-represented categories also included other families such as Glutamate receptors and G-protein modulators (*Figure 6C*).

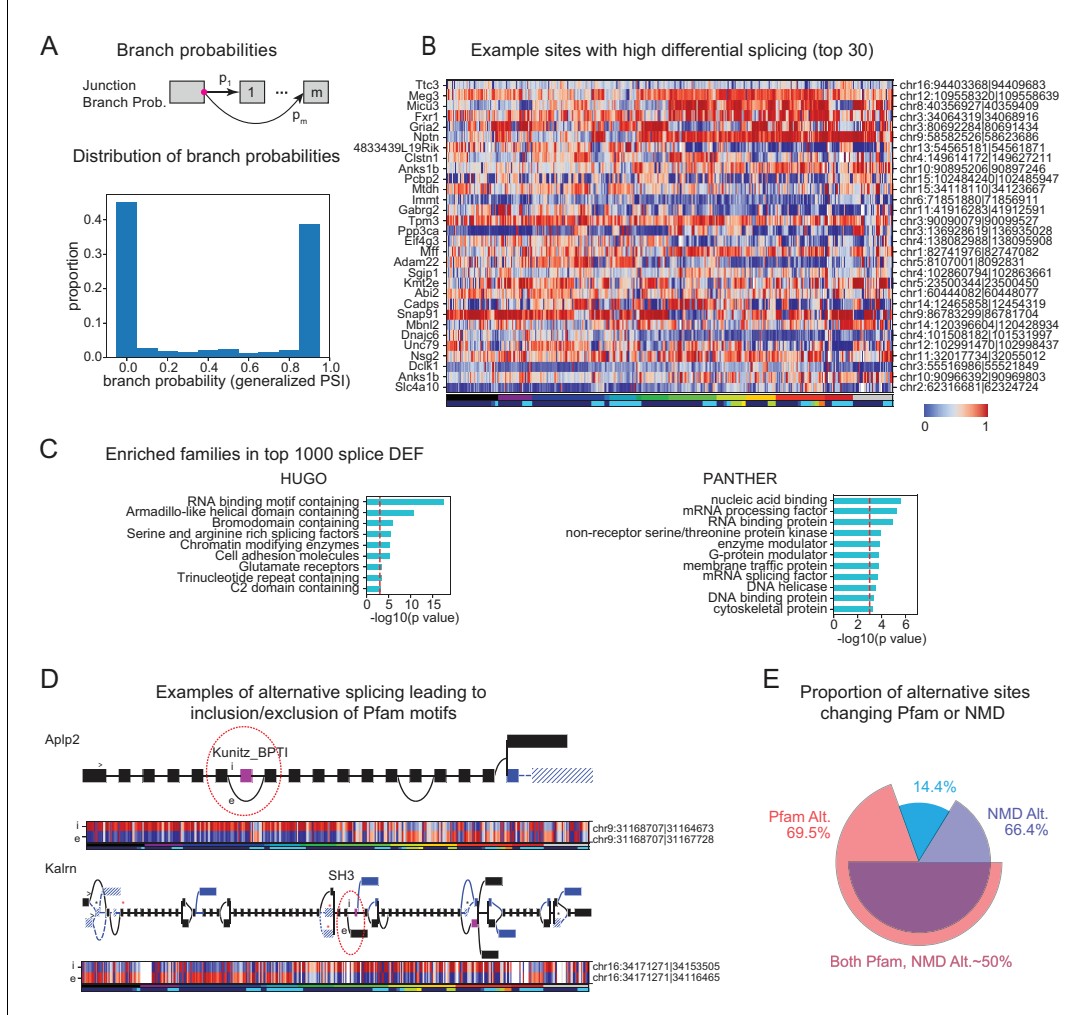

**Figure 6.** Alternative splicing and neuronal diversity. (A) (Top) Schematic representation of branch probabilities. Alternative donor sites (red dot) can be spliced to multiple acceptor sites $1, \ldots, m$ with probabilities $p_1, \ldots, p_m$. (Bottom) Distribution of branch probabilities across all samples and all alternative splice sites. (B) Heatmap showing branch probabilities across neuronal samples for branches with highest splice DEF. Each row corresponds to a branch within the indicated gene on the left and the location is indicated on the right. Samples without junctional reads at this branch are colored white. (C) Enriched HUGO gene groups and PANTHER protein classes for genes with top 1000 combined splice DEF. (D) Splice graphs illustrating examples of alternative splicing leading to inclusion or exclusion (marked 'i', 'e' of Pfam domains (magenta exons) with branch probabilities shown in the heatmap below. Previously unannotated exons and junctions are blue; annotated are black. Dotted lines indicate branches predicted to lead to nonsense-mediated decay (NMD). A red star above an exon indicates existence of a premature termination codon (PTC) within the exon which satisfies the '50nt rule' for NMD (*Nagy and Maquat, 1998*) (i.e. more than 50 bp upstream to the next junction), whereas a black star indicates existence of a PTC within 50 bp of the next junction. Dashed lines and hatches indicate that there is no coding path through the element. (>) indicates an annotated translation start site. (E) Proportion of branch points predicted to lead to NMD (purple), altered Pfam inclusion (red), or both (overlapped region), at one or more of its branches.

DOI: https://doi.org/10.7554/eLife.38619.024

To begin to assess the functional impact of alternative splicing, we determined which alternative sites lead to inclusion or exclusion of a known protein domain using the Pfam database (*Finn et al., 2015*). In addition to providing information relevant to the potential functions of many previously unknown isoforms, our analysis also provides a more comprehensive view of known splice events. Two examples are shown in *Figure 6D*. Alternative splicing of Amyloid precursor-like protein 2 (*Aplp2*) is known to regulate inclusion of a bovine pancreatic trypsin inhibitor (BPTI) Kunitz domain (*Sandbrink et al., 1997*) and this domain is known to regulate proteolysis of the related protein APP, the amyloid precursor protein implicated in Alzheimer's disease (*Beckmann et al., 2016*). Differential inclusion of this exon is known to occur between neurons and nonneurons. Intriguingly, we

found that splicing at this site in hippocampal interneurons differs not only from that in forebrain excitatory neurons, but also from other forebrain inhibitory neurons in neocortex and striatum. Kalirin (*Kalrn*) is a RhoGEF kinase implicated in Huntington's disease, schizophrenia and synaptic plasticity (*Penzes and Jones, 2008*). Kalrn is known to be regulated via binding of adaptor proteins to its SH3 (SRC homology 3) domains (*Schiller et al., 2006*) which is regulated by alternative splicing of this domain. In addition to expanding the number of known variants (blue exons and junctions in *Figure 6D*) we reveal their detailed distribution across the profiled set of neural populations. In total, the data reveal a detailed quantitative view of hundreds of thousands of known and unknown cell-type-specific splicing events, providing an unmatched resource for investigating their functional significance.

Not all splicing events alter the inclusion or exclusion of known protein domains. Many splicing events introduce frame shifts or new stop codons and hence are predicted to lead to nonsense-mediated decay (NMD). Coupling of regulated splicing to NMD is believed to be an important mechanism for regulating protein abundance (*Lewis et al., 2003*). Consistent with previous observations (*Yan et al., 2015*; *Mauger and Scheiffele, 2017*), we noticed that most alternative sites contain branches that can lead to NMD (*Figure 6E*). This suggests that alternative splicing may contribute not only to the diversity of isoforms present, but also to diversity defined on the basis of transcript abundance.

The present results provide a comprehensive resource of known and novel splicing events across a large number of neuronal cell types. Altogether, nearly 70% of alternative sites lead to differential inclusion of a known Pfam domain or NMD (*Figure 6E*), and thus to functional or quantitative diversity across cell types.

## Long genes contribute disproportionately to neuronal diversity

We found that neuronal effector genes (ion channels, receptors and cell adhesion molecules, etc.) have the greatest ability to distinguish cell populations (*Figure 4E*). Previously, these categories of genes have been found to be selectively enriched in neurons and to share the physical characteristic of being long (*Sugino et al., 2014*; *Gabel et al., 2015*; *Zylka et al., 2015*). Consistent with this, DEF, which approximates the mutual information (MI) between expression levels and cell populations, is significantly correlated with length (*Figure 7A*; correlation coefficient = 0.19; p=7.5e-189), reaching a maximum for the very longest genes. Long genes ($\geq$100 kb) have nearly twice the average ability to distinguish cell populations (DEF) as shorter genes (*Figure 7A*), and provide greater family-wise separation between cell types (*Figure 7C*). Analyzing publicly available single-cell data confirms that this bias is broadly observable (*Figure 7—figure supplement 1*). In contrast, FCR, which measures the signal-to-noise or robustness of expression differences, is higher for shorter genes, reaching a maximum for genes below 10 kbp in length (*Figure 7B*).

Recently, (*Raman et al., 2018*) have argued that many prior observations of long gene bias are not significant when controlling for baseline variability in length-dependent expression. In order to assess the applicability of this argument to the present observations, we compared the fold-changes across length between groups and within replicates of individual groups as in *Raman et al. (2018)*. An example of this test applied to two populations is shown in *Figure 7—figure supplement 2A,B*. Even after applying corrections for multiple comparisons across all bins, the long gene bins ($\geq$100 kb) are highly significant. Panels C,D of this figure illustrate the results of performing this comparison for all GACPs in our dataset. The median fraction of significant long gene bins (0.89) greatly exceeded the fraction of short gene bins (0.1). A more detailed analysis of the test developed by Raman et al. and its application to other observations will be published elsewhere.

In addition to being differentially expressed, long genes are likely to have a larger number of exons and hence a greater potential for differential splicing. To evaluate the degree to which differential splicing of long genes contributes to distinguishing cell populations we plotted the splice DEF (*Figure 6*) as a function of gene length. As expected, DEF calculated from differential splicing also increased with gene length (*Figure 7D*), although the slope was more gradual and the maximum DEF value achieved was less than that for gene expression (*Figure 7A*). For each gene, we measured the fraction of cell populations pairs that could be distinguished on the basis of differential expression, differential splicing, or both. This revealed that for the current dataset, the average alternatively spliced gene distinguishes only 1.4% of cell populations, but distinctions based on expression of these same genes were nearly 10 times more common (13.9%, *Figure 7E*).

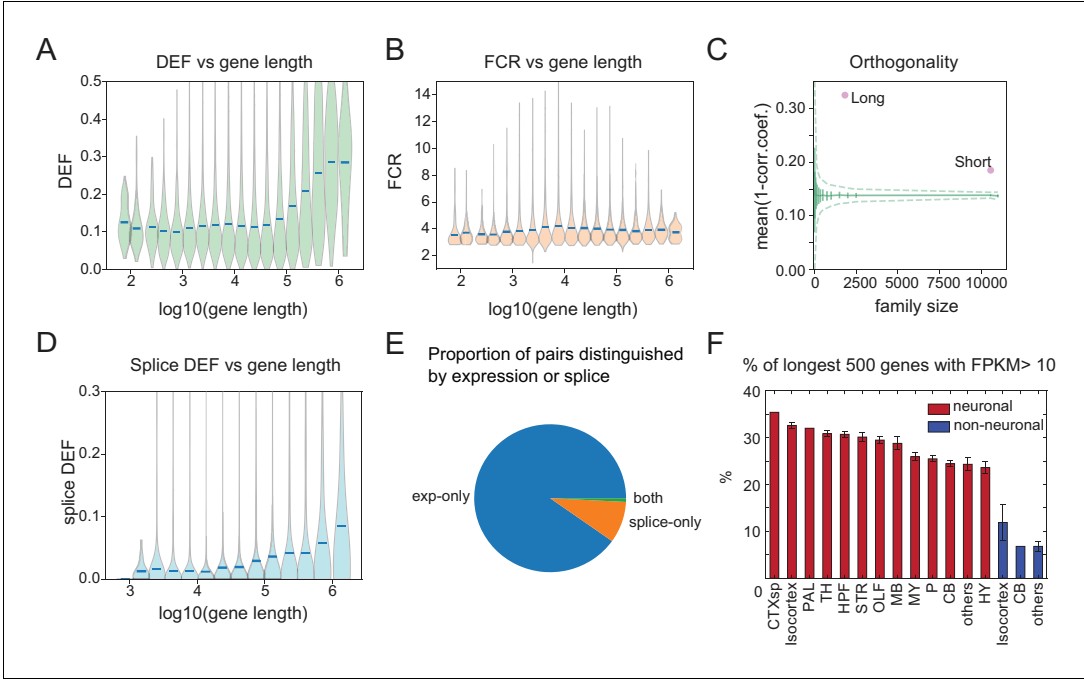

**Figure 7.** Long genes have a greater capacity for distinguishing cell populations. (**A**) DEF as a function of gene length. For violin plots in A, B, D, genes are sorted by length and binned (four bins per log unit). (**B**) Robustness of expression difference (FCR) as a function of gene length. (**C**) Orthogonality of expression patterns calculated as in *Figure 5E*, but using long neuronal genes (n = 1829, ≥100 kb) and short neuronal genes (n = 10572, <100 kb) rather than functionally defined gene families. Z-score is 33.2 for long and 22.1 for short neuronal genes. Both are highly different from randomly sampled genes (green solid lines mean and Std. Dev.; dashed lines = 99% confidence interval), but long genes provide greater separation. (**D**) Splice DEF as a function of gene length. (**E**) Fraction of pairs distinguished by splicing (splice-only), transcript abundance (exp-only), or by both measures. (**F**) Variation in long gene expression in neuronal and nonneuronal populations across major brain regions studied. Error bars are SEM. (CTXsp consisted of single region: Claustrum.)
DOI: https://doi.org/10.7554/eLife.38619.025

The following figure supplements are available for figure 7:

**Figure supplement 1.** DEF length bias in SC datasets.
DOI: https://doi.org/10.7554/eLife.38619.026
**Figure supplement 2.** Significant length differences using the test proposed by *Raman et al. (2018)* evaluating length dependent differences by comparing expression ratios between groups to those within a single group.
DOI: https://doi.org/10.7554/eLife.38619.027
**Figure supplement 3.** Regional bias of long gene expression in SC datasets.
DOI: https://doi.org/10.7554/eLife.38619.028

Finally, to determine whether neuronal long gene expression contributes more to profiles in some anatomical regions than in others, we plotted the fraction of the longest genes expressed in neuronal and nonneuronal populations across each of the major brain regions studied. The results confirm strong differences between neurons and nonneurons and show the strongest long gene expression in forebrain regions, with weaker expression evident in hindbrain (*Figure 7F*). Analyses of single-cell datasets revealed similar trends (*Figure 7—figure supplement 3*).

## Discussion

### A resource of genetically identified neuronal transcriptomes

The dataset presented here is the largest collection of transcriptomes of anatomically and genetically specified neuronal cell types available in a mammalian species (*Table 1*). The approach employed in this study provides a complementary view of neuronal diversity to that afforded by SC

sequencing. By sorting and pooling ~100 cells chosen based on genetic and anatomical similarity, we generated profiles with low noise and high depth, but, where tested, with a comparable degree of homogeneity, as that obtained in recent SC studies.

The fact that each transcriptome corresponds to a genetically (or retrogradely) labeled population will foster reproducible studies across investigators. The few profiles in our study that mapped to more than one SC profile (*Figure 2*), may represent cell types better distinguishable using SCs or improved genetic markers, or alternatively, may represent cell populations that are highly overlapping. The optimal granularity with which cell types may be distinguished remains an open question. Pooling cell profiles either prior to sequencing, as in this study, or after sequencing at the clustering phase, as in SC studies, risks compromising profile homogeneity. However, over-fragmenting clusters risks the opposite problem of reducing the reliability and reproducibility with which populations can be distinguished across studies. Given the complementary advantages of improved reproducibility and separability afforded by pooling profiles, and of reduced heterogeneity afforded by maximally separating profiles, further integration of these approaches with other modalities, such as FISH (*Moffitt et al., 2016*) are needed to accurately profile the full census of brain cell types. By linking these efforts to genetically identified neurons, the present dataset provides a useful resource for these efforts.

## A transcriptional code for neuronal diversity

We utilized easily calculated metrics that capture essential features of the robustness and information content of transcriptome diversity. These measures are simply versions of Fold-Change (FCR) and Differential Expression (DEF) adapted to the analysis of many separate populations simultaneously. Importantly, they capture independent components of the differences captured by variance-based metrics like ANOVA and CV (*Figure 4A*, *Figure 3—figure supplement 1*). Metrics like ANOVA are influenced jointly by signal-to-noise and mutual information, while FCR and DEF better separate them (*Figure 3—figure supplement 1*) and so these metrics may be more broadly useful when making genome-wide comparisons across many populations. In the present dataset, FCR and DEF identified two very different sets of genes contributing to neuronal diversity: high FCR, low-noise genes, exemplified by homeobox transcription factors, and high DEF, long neuronal effector genes like ion channels, receptors and cell adhesion molecules.

The homeobox family of TFs exhibited the most robust (high FCR) expression differences across cell types (*Figure 4D*). These ON/OFF differences were characterized by extremely low expression in the OFF state (*Figure 5*). Mechanistically, the low expression was associated with reduced genome accessibility measured by ATAC-seq (*Figure 5C,D*), presumably reflecting epigenetic regulation of the OFF state, known to occur for example at the clustered Hox genes via Polycomb group (PcG) proteins (*Montavon and Soshnikova, 2014*). Although this regulation has been studied most extensively at Hox genes, genome-wide ChIP studies reveal that PcG proteins are bound to over 100 homeobox TFs in ES cells (*Boyer et al., 2006*). Our results indicate that strong cell-type-specific repression persists in the adult brain, presumably due to the continued functional importance of preventing even partial activation of inappropriate programs of neuronal identity.

Although individually, homeobox TFs contain less information about cell types than long neuronal effector genes, their patterns of expression are highly orthogonal and therefore their joint expression pattern is highly informative. As a group, homeobox TFs distinguished more than 99% of neuronal cell types profiled (*Figure 5—figure supplement 2*). (Note this includes several Purkinje and hippocampal pyramidal cell groups that may actually represent duplicate examples of the same cell types). Historically, homeobox TFs are well known to combinatorially regulate neuronal identity in *Drosophila* and *C. elegans* (*Pereira et al., 2015*; *Gendrel et al., 2016*) and the vertebrate brainstem and spinal cord (*Dasen and Jessell, 2009*; *Philippidou and Dasen, 2013*). Our results suggest a broader importance of homeobox TFs throughout the mammalian nervous system. Continued expression of these factors in adult neurons suggests that they likely also contribute to the maintenance of neuronal identity.

## Long genes and neuronal diversity

Our study suggests that long neuronal effector genes contribute disproportionately to neuronal transcriptional diversity (*Figure 7*). Previously, it was reported that differences in transcript length can

bias differential expression analysis of RNA-seq data (*Oshlack and Wakefield, 2009*). To ensure that we avoided this bias, we used counts of reads only from within the one kbp-long 3' ends of the genes for calculating expression values. Recently, an alternative statistical analysis has been used to argue that some of these length biases may be artefactual (*Raman et al., 2018*). Despite concerns about the rigor of this analysis (manuscript in preparation), we found that the observed length biases remain highly significant, even within this statistical framework (*Figure 7—figure supplement 2*), suggesting that they are robust features of the transcriptional differences between neuronal populations.

Long genes are expressed at higher levels in neurons than in nonneuronal cells in the nervous system, a bias that was also present in SC datasets (*Figure 7—figure supplements 1* and *2*) and that has been reported previously (*Sugino et al., 2014*; *Gabel et al., 2015*; *Zylka et al., 2015*). These differences are greatest in the forebrain (*Figure 7F*; *Figure 7—figure supplement 2*), perhaps reflecting the large numbers of distinct cell types in these regions and the enhanced ability of these genes to distinguish GACPs based on their expression. However, we and others did not measure cell-type-specific protein expression, and so cannot be sure that the long gene bias extends to the level of neuronal proteins.

Long genes tend to have larger numbers of exons and therefore are likely to be expressed in a larger number of distinct isoforms as a result of alternative splicing (alternative start sites also contribute). We quantified differential splicing from analysis of junctional reads. Interestingly, branch probabilities at most sites of alternative splicing were highly bimodal (*Figure 6A*), suggesting that within each GACP, splicing is largely all or none, a finding previously reported in single immune cells (*Shalek et al., 2013*) but not found in some single neuron studies (*Gokce et al., 2016*). This led to patterns that often flipped between high and low probabilities for a given branch as one traversed major brain region boundaries (*Figure 6B*). More than two thirds of these splicing events lead to inclusion or exclusion of known protein domains (*Figure 6E*), but many of these, as well as some of the remaining events that do not modify domain structure, also introduce a frame shift or premature stop codon, and so are predicted to lead to nonsense mediated decay (NMD). We did not directly test the contribution of NMD to transcript abundance, but our splicing results are consistent with the idea that this may be an important mechanism for regulating transcript stability and hence transcript abundance across different cell populations (*Yan et al., 2015*; *Traunmüller et al., 2014*). While differential splicing is able to distinguish fewer GACPs than transcript abundance (*Figure 7E*), this may be an underestimate for two reasons. First, as just noted, splicing may influence transcript abundance through NMD, and second, the sensitivity to detect splicing differences depends on an adequate number of junctional reads. Deeper sequencing could increase the apparent contribution of this component of neuronal diversity.

Long genes are enriched in the signaling molecules, receptors and ion channels responsible for input/output transformations in neurons, and the cell adhesion molecules that specify neuronal connectivity. The finding that these genes play an important role in diversifying cortical interneurons (*Paul et al., 2017*), as well as distinguishing the larger set of populations studied here, is sensible in light of the phenotypic diversity required for neuronal communication and connectivity. These genes are long because of long introns that are rich in sequences derived from transposons and other retroelements (*Grishkevich and Yanai, 2014*). Whether or how this increased length has any functional significance for the regulation of these genes is unclear from our studies, but it is intriguing that these long genes are disrupted in forms of autism spectrum disorder (*Zylka et al., 2015*; *Wei et al., 2016*) and in the related developmental disorder Rett Syndrome (*Sugino et al., 2014*; *Gabel et al., 2015*), where loss of the chromatin protein Mecp2 leads to selective upregulation of long neuronal genes in a highly cell-type-specific fashion. These studies suggest the possibility that long neuronal genes are subject to distinct modes of regulation, with particular significance for neuronal diversity.

In contrast to long neuronal effector genes, which tend to be expressed later in development as neurons mature phenotypically (*Okaty et al., 2009*), low noise, high FCR genes are frequently critical for early development. These genes, such as many of the homeobox TFs, are often quite short and, at least in the case of the Hox genes, are known to be remarkably transposon impoverished (*Waterston et al., 2002*; *Simons et al., 2006*). This may reflect selection against transposon insertion, but may also reflect chromatin that is non-permissive for insertion in germ cells and the early embryo, where heritable transposition occurs. The high FCR/low OFF noise of many of these genes detected here may reflect a transcriptional signature of this class of genes. Consistent with this view,

low OFF noise genes were nearly six times shorter than high OFF noise genes (*Figure 5—figure supplement 1D*). Highly restrictive chromatin at these genes may be established early in development to protect them from disruptive transposition (*Montavon and Soshnikova, 2014*). If so, this tightly closed state is maintained in postmitotic neurons where it may also prevent transcriptional signals associated with inappropriate neural identities. This feature was not uniformly present across all subfamilies of homeobox transcription factors. Interestingly, however, the families with the highest FCR and lowest noise also had the shortest length, while those with higher noise expression (and lower FCR) were longer (*Figure 5—figure supplement 4*).

The observation that long genes contribute disproportionately to neuronal transcriptional diversity is surprising both because of the increased metabolic cost of expressing them (*Castillo-Davis et al., 2002*), and since these genes are frequent sites of genome instability associated with genetic lesions leading to autism and other developmental disorders (*Wei et al., 2016*). These apparent disadvantages may be too weak to lead to selection against long gene expression in mammalian neurons. If this is not the case, however, it raises the question of why the mechanisms used to prevent elongation of shorter, low OFF noise genes were not also applied to neuronal effector genes. This could simply reflect developmental or later functional constraints that exclude the use of these epigenetic protection mechanisms. Alternatively, length itself may confer some advantages that outweigh other disadvantages. This could occur either through benefits provided by the diversification of alternative splicing, or through regulatory features contained within intronic sequences (*Zhao et al., 2018*).

# Materials and methods

## Cell types and mouse lines

We assume that cell types are organized hierarchically in a tree-like fashion proceeding from major branches (e.g. 'cortical excitatory neuron') to more specialized subtypes, with the terminal 'leaf-level' branches comprising 'atomic' cell types. Profiled cell populations are defined operationally by the intersection of a transgenic mouse strain (or in some cases anatomical projection target) and a brain region. Mouse lines profiled in this study are summarized in *Supplementary file 1*. Most were obtained from GENSAT (*Gong et al., 2007*) or from the Brandeis Enhancer Trap Collection (*Shima et al., 2016*). For Cre-driver lines, the Ai3, Ai9 or Ai14 reporter (*Madisen et al., 2010*) was crossed and offspring hemizygous for Cre and the reporter gene were used for profiling. Information on samples profiled is in *Supplementary file 2*. Populations profiled are designed to sample regions and cell types across the mouse brain within the limits of available resources. In addition several non-brain samples were profiled as out-groups. Replicate numbers (averaging three across all populations) are in *Supplementary file 2*. Replicates were obtained in single animals, except for a few cases in which pooling across animals was needed due to difficulty in sorting. Our study used a small number of replicates (n = 2–4) per population to maximize the number of populations studied, while still allowing calculation of summary statistics. No explicit power analysis was performed. No attempt was made to remove outliers. Sequenced libraries were not used when total reads were low (<5M reads). Out of 179 neuronal GACPs, there are 165 groups which have more than one replicate. Of these, 14 were recent additions, and most analyses were performed with the remaining 151 groups. All experiments were conducted in accordance with the requirements of the Institutional Animal Care and Use Committees at Janelia Research Campus and Brandeis University.

## Tissue data

In addition to cell-type-specific data obtained in this study, we analyzed publicly available RNA-seq data using tissue samples. Information on these samples are described in *Supplementary file 3*.

## Atlas

Animals were anesthetized and perfused with 4% paraformaldehyde and brains were sectioned at $50\mu m$ thickness. Every fourth section was mounted on slides and imaged with a slide scanner equipped with a 20x objective lens (3DHISTECH; Budapest, Hungary). In house programs were used to adjust contrast and remove shading caused by uneven lighting. Images were converted to a zoomify-compatible format for web delivery and are available at http://neuroseq.janelia.org.

## Cell sorting

Manual cell sorting was performed as described (*Hempel et al., 2007*; *Sugino et al., 2014*). Briefly, animals were sacrificed following isoflurane anesthesia, and $300\mu m$ slices were digested with pronase E (1 mg/ml, P5147; Sigma-Aldrich) for 1 hr at room temperature, in artificial cerebrospinal fluid (ACSF) containing 6,7-dinitroquinoxaline-2,3-dione ($20\mu M$; Sigma-Aldrich), D-$(-)-$2-amino-5-phosphonovaleric acid ($50\mu M$; Sigma-Aldrich), and tetrodotoxin ($0.1\mu M$; Alomone Labs). Desired brain regions were micro-dissected and triturated with Pasteur pipettes of decreasing tip size. Dissociated cell suspensions were diluted 5–20 fold with filtered ACSF containing fetal bovine serum (1%; HyClone) and poured over Petri dishes coated with Sylgard (Dow Corning). For dim cells, Petri dishes with glass bottoms were used. Fluorescent cells were aspirated into a micropipette (tip diameter 30–50$\mu m$) under a fluorescent stereomicroscope (M165FC; Leica), and were washed three times by transferring to clean dishes. After the final wash, pure samples were aspirated in a small volume ($1\sim3\mu l$) and lysed in $47\mu l$ XB lysis buffer (Picopure Kit, KIT0204; ThermoFisher) in a $200\mu l$ PCR tube (Axygen), incubated for 30 min at 40℃ on a thermal cycler and then stored at $-80$℃. Detailed information on profiled samples are provided in *Supplementary file 2*.

## RNA-seq

Total RNA was extracted using the Picopure kit (KIT0204; ThermoFisher). Either 1 $\mu l$ total, or $1\mu l$ per 50 sorted cells of $10^{-5}$ dilution of ERCC spike-in control (#4456740; Life Technologies) was added to the purified RNA and vacuum concentrated to 5 $\mu l$ and immediately processed for reverse transcription using the NuGEN Ovation RNA-Seq System V2 (#7102; NuGEN) which yielded $4\sim8\mu g$ of amplified DNA. Amplified DNA was fragmented (Covaris E220) to an average of ~200 bp and ligated to Illumina sequencing adaptors with the Encore Rapid Kit (0314; NuGEN). Libraries were quantified with a KAPA Library Quant Kit (KAPA Biosystems) and sequenced on an Illumina HiSeq 2500 with 4 to 32-fold multiplexing (single end, usually 100 bp read length, see *Supplementary file 2*).

## RNA-seq analysis

Adaptor sequences (AGATCGGAAGAGCACACGTCTGAACTCCAGTCAC for Illumina sequencing and CTTTGTGTTTGA for NuGEN SPIA) were removed from de-multiplexed FASTQ data using cutadapt v1.7.1 (http://dx.doi.org/10.14806/ej.17.1.200) with parameters '–overlap = 7 –minimumlength = 30'. Abundant sequences (ribosomal RNA, mitochondrial, Illumina phiX and low complexity sequences) were detected using bowtie2 (*Langmead and Salzberg, 2012*) v2.1.0 with default parameters. The remaining reads were mapped to the UCSC mm10 genome using STAR (*Dobin et al., 2013*) v2.4.0i with parameters '–chimSegmentMin 15 –outFilterMismatchNmax 3'. Mapped reads are quantified with HTSeq (*Anders et al., 2015*) using Gencode.vM13 (*Harrow et al., 2012*).

## Annotations

For reference annotations we used Gencode.vM13 (*Harrow et al., 2012*) downloaded from http://www.gencodegenes.org/, and NCBI RefSeq (*Pruitt et al., 2014*) downloaded from the UCSC genome browser.

## Pan-neuronal genes

Pan-neuronal genes satisfied the following conditions: (1) mean neuronal expression level (NE) > 20 FPKM, (2) minimum NE > 5 FPKM, (3) mean NE > maximum nonneuronal expression level (NNE), (4) minimum NE > mean NNE, (5) mean NE > 4x mean NNE, (6) mean NE > mean NNE + 2x standard deviation of NNE, (7) mean NE $-$ 2x standard deviation of NE > mean NNE.

## DEF/FCR/DM calculation

To calculate DEF, the following criteria were used to assign a '1' or '0' to each element in the differentiation matrix (DM): absolute log fold change >2 and q-value <0.05. Q-values were calculated using the limma package including the voom method (*Law et al., 2014*). To adjust the power to be similar across cell types, two replicates (the most recent two) were used for all cell populations with more than two replicates. We have tried the same calculations with three replicates (using a fewer number of cell populations) and obtained similar results (data not shown). To avoid possible bias in

variances due to transcript length differences (*Oshlack and Wakefield, 2009*), we quantified counts using reads from within the 3' 1 kbp of each gene. For genes with transcript lengths shorter than one kbp, we used the whole gene length. We also calculated DEF and FCR across five SC datasets: For (*Zeisel et al., 2015*; *Tasic et al., 2016*) and (*Tasic et al., 2018*), we used log fold change >1 and q-value <0.05 calculated using the limma/voom method for differential gene expression. For (*Saunders et al., 2018*) and (*Zeisel et al., 2018*), only cluster average expression was available, and log fold change >1 was defined as the criterion for differential expression.

## Overrepresentation, orthogonality and minimal gene sets

Overrepresentation analysis was performed using the top-level HUGO gene groups (*Figures 4–6*) and was supplemented (*Figure 6*, *Figure 4—figure supplement 1*, *Figure 5—figure supplements 1* and *3*) using the PANTHER Classification System and the Molecular Function component of the Gene Ontology Annotation (GOM). Orthogonality quantifies the non-redundancy across expression patterns. We calculated orthogonality (*Figure 5E*) as the mean pairwise decorrelation (1- Pearson's corr. coef.) over a family of genes. Gene groups with less than 50 members were excluded, since variance of this measure was much larger in small groups of randomly selected genes (dashed green lines in *Figure 5E*). Minimal gene sets capable of serving as combinatorial codes across cell populations (*Figure 5—figure supplement 2*) were calculated by a greedy algorithm using the Differentiation Matrix (DM) defined in *Figure 3*. Specifically, from a set of genes (such as homeobox TFs or other families), the gene with the highest DEF was chosen as the first member of the set. Successive members were chosen, irrespective of their individual DEF, so as to maximize the combined DEF of the set. The combined DEF is the fraction of pairs distinguished by any gene in the group, and is calculated from the combined DM, which is the logical OR of the individual DMs for each gene in the group. This procedure continued until the combined DEF exceeded the desired threshold (0.99 in the case of *Figure 5—figure supplement 2*). The homeoboxes set was constructed by merging the HUGO Homeoboxes gene group and the PANTHER homeobox protein TFs (PC00119) and had 156 genes. The GPCRs set is a merging of G protein-coupled receptors in HUGO and G-protein coupled receptors (PC00021) in PANTHER and has 347 genes.

## Calculation of differential splicing

To identify differential splicing, we utililzed a statistical test based on the Dirichlet-Multinomial distribution and the log-likelihood ratio test, developed in LeafCutter (*Li et al., 2018*). However, instead of using a group of connected introns as a unit for tests (as done in LeafCutter), we used a group of introns originating from an alternative donor site. Total junctional reads at an alternative donor > 10 was a prerequisite for testing. DM for alternative donors were then calculated as 1 for pairs of cell populations with $p<0.05$ and maximum delta-PSI > 0.1, and 0 for others. (delta-PSI: absolute difference of PSI, proportion-spliced-in,)

## NNLS/random forest decomposition

The following single-cell datasets were downloaded and used for decomposition: (*Zeisel et al., 2015*) (NCBI GEO GSE60361), (*Tasic et al., 2016*) (NCBI GEO GSE71585), (*Tasic et al., 2018*) (http://celltypes.brain-map.org/rnaseq), (*Zeisel et al., 2018*) (http://mousebrain.org/), (*Saunders et al., 2018*) (dropviz.org). Deposited count data were converted to $log_2(CPM+1)$ and used for comparison. The NeuroSeq dataset was quantified using RefSeq and featurecount (*Liao et al., 2014*) and converted into $log_2(CPM+1)$. Subsets of genes common to NeuroSeq, Tasic 2018 and Zeisel 2018 datasets were used for decomposition. To account for differences in distributions of logCPM values between datasets, they were quantile-normalized to an average profile generated from the decomposed dataset. Since most genes in the single-cell profiles exhibited noisy expression patterns, using the entire gene set for decomposition was not feasible. Therefore, we selected genes deemed most informative for distinguishing cell classes based on the ANOVA F-statistic across cell classes (obtained using limma/voom in R). However, simply taking the top ANOVA genes led to highly biased gene selection since some cell types exhibited much larger transcriptional differences than others (e.g. many ANOVA selected genes were specific to microglia). We therefore selected genes to reduce the redundancy between distinguished cell populations. Beginning with the highest ANOVA gene (highest ANOVA F-value), genes were selected only if their DM

(Differentiation Matrix defined in *Figure 3*) differed from those previously selected, enforced by requiring a Jaccard index threshold of 0.5, across all studies. We chose the top 500 genes meeting this criterion. Decompositions were performed on average profiles created by averaging NeuroSeq replicates or by averaging single-cell profiles using cluster assignments provided by the authors. NNLS was implemented using the R nnls library. For Random forest, the randomForest R package was used.

## ATAC-seq

Seven cell types, Purkinje and granule cells from cerebellum, pyramidal cells in layer 5 and 6 from neocortex, in the deep layers of entorhinal cortex, and in CA1 and CA1-3 of hippocampus, labeled in mouse lines P036, P033, P078, 56L, P038, P064, and P036 respectively (all from *Shima et al., 2016*) were profiled with ATAC-seq. They were isolated by FACS to obtain ~40,000 labeled neurons. ATAC libraries for Illumina next-generation sequencing were prepared in accordance with a published protocol (*Buenrostro et al., 2013*). Briefly, collected cells were lysed in buffer containing 0.1% IGEPAL CA-630 (I8896, Sigma-Aldrich) and nuclei pelleted for resuspension in tagmentation DNA buffer with Tn5 (FC-121–1030, Illumina). Nuclei were incubated for 20–30 min at 37˚C. Library amplification was monitored by real-time PCR and stopped prior to saturation (typically 8–10 cycles). Library quality was assessed prior to sequencing using BioAnalyzer estimates of fragment size distributions looking for a ladder pattern indicative of fragmentation at nucleosome intervals as well as qPCR to determine relative enrichment at two housekeeping genes compared to background (specifically the TSS of *Gapdh* and *Actb* were assessed relative to the average of three intergenic regions). For sequencing, Illumina HiSeq 2500 with 2 to 4-fold multiplexing and paired end 100 bp read length was used. In addition to ATAC-seq, RNA-seq was performed on replicate samples of ~2000 cells collected in a similar way, and library prepared using the same method described above.

## ATAC-seq analysis

Nextera adaptors (CTGTCTCTTATACACATCT) were trimmed from both ends from de-multiplexed FASTQ files using cutadapt with parameters "-n 3 -q 30,30 m 36'. Reads were then mapped to UCSC mm10 genome using bowtie2 (*Langmead and Salzberg, 2012*) with parameters "-X2000 –no-mixed –no-discordant'. PCR duplicates were removed using Picard tools (http://broadinstitute.github.io/picard, v2.8.1) and reads mapping to mitochondrial DNA, scaffolds, and alternate loci were discarded. BigWig genomic coverage files were generated using bedtools (*Quinlan and Hall, 2010*) and scaled by the total number of reads per million.

## Anatomical region abbreviations

Region abbreviations: ACB: Nucleus accumbens AD: Anterodorsal nucleus AI: Agranular insular area AMd: Anteromedial nucleus, dorsal part AOBgr: Accessory olfactory bulb, granular layer AOBmi: Accessory olfactory bulb, mitral layer AP: Area postrema ARH: Arcuate hypothalamic nucleus AV: Anteroventral nucleus of thalamus CA: Hippocampus Ammon's horn CA1: Hippocampus field CA1 CA1sp: Hippocampus field CA1, pyramidal layer CA3: Hippocampus field CA3 CEAm: Central amygdalar nucleus, medial part CEAl: Central amygdalar nucleus, lateral part CL: Central lateral nucleus of the thalamus COAp: Cortical amygdalar area, posterior part CP: Caudoputamen CSm: Superior central nucleus raphe, medial part CUL4,5gr: Cerebellum lobules IV-V, granular layer CUL4,5mo: Cerebellum lobules IV-V, molecular layer CUL4,5pu: Cerebellum lobules IV-V, Purkinje layer DCO: Dorsal cochlear nucleus DG: Hippocampus dentate gyrus DMHp: Dorsomedial nucleus of the hypothalamus, posterior part DMX: Dorsal motor nucleus of the vagus nerve DR: Dorsal nucleus raphe ECT: Ectorhinal area IC: Inferior colliculus IG: Induseum griseum IO: Inferior olivary complex isl: Islands of Calleja islm: Major island of Calleja LC: Locus ceruleus LGd: Dorsal part of the lateral geniculate complex LHA: Lateral hypothalamic area MM, Medial mammillary nucleus MO: Somatomotor area MOBgl: Main olfactory bulb, glomerular layer MOBgr: Main olfactory bulb, granular layer MOBmi: Main olfactory bulb, mitral layer MOE: main olfactory epithelium MOp5: Primary motor area, layer 5 MV: Medial vestibular nucleus NTS: Nucleus of the solitary tract NTSge: Nucleus of the solitary tract, gelatinous part NTSm: Nucleus of the solitary tract, medial part ORBm: Orbital area, medial part OT: Olfactory tubercle PAG: Periaqueductal gray PBl: Parabrachial nucleus, lateral

division PCN: Paracentral nucleus PG: Pontine gray PIR: Piriform area PRP: Nucleus prepositus PVH, Paraventricular hypothalamic nucleus PVHd: Paraventricular hypothalamic nucleus, descending division PVHp, Paraventricular hypothalamic nucleus, parvicellular division PVT: Paraventricular nucleus of the thalamus PYRpu: Cerebellum Pyramus (VIII), Purkinje layer RPA: Nucleus raphe pallidus RSPv: Retrosplenial area, ventral part RT, Reticular nucleus of the thalamus SCH: Suprachiasmatic nucleus SCm: Superior colliculus, motor related SFO: Subfornical organ SNc: Substantia nigra, compact part SO: Supraoptic nucleus SSp: Primary somatosensory area SSs: Supplemental somatosensory area SUBd-sp: Subiculum, dorsal part, pyramidal layer VII: Facial motor nucleus VISp: Primary visual area VISp6a: Primary visual area, layer 6a VNO: vemoronasal organ VPM: Ventral posteromedial nucleus of the thalamus VTA: Ventral tegmental area

## Acknowledgements

We thank Jody Clements and Charlotte Weaver for help in preparing the web site, Erina Hara, Asish Gulati, Xiaotang Jing and Zhe Meng for technical help, Keven McGowan for assistance in sequencing, Jim Cox, Amanda Zeladonis and Amanda Wardlaw for help in animal maintenance, Gabe Murphy for help in retinal sample collection, and Rosa Miyares for comments on the manuscript.

## Additional information

### Funding

| Funder | Grant reference number | Author |
|---|---|---|
| Howard Hughes Medical Institute | | Ken Sugino<br>Anton Schulmann<br>Lihua Wang<br>David L Hunt<br>Bryan M Hooks<br>Dimitri Tränkner<br>Jayaram Chandrashekar<br>Andrew L Lemire<br>Nelson Spruston<br>Adam W Hantman<br>Sacha B Nelson |
| National Eye Institute | EY022360 | Erin Clark |
| National Institute of Mental Health | MH105949 | Erin Clark<br>Yasuyuki Shima<br>Sacha B Nelson |
| National Institute of Neurological Disorders and Stroke | NS075007 | Erin Clark<br>Yasuyuki Shima<br>Sacha B Nelson |

The funders had no role in study design, data collection and interpretation, or the decision to submit the work for publication.

### Author contributions

Ken Sugino, Conceptualization, Resources, Data curation, Software, Formal analysis, Supervision, Investigation, Visualization, Methodology, Writing—original draft, Writing—review and editing; Erin Clark, Yasuyuki Shima, Bryan M Hooks, Resources, Investigation, Writing—review and editing; Anton Schulmann, Software, Formal analysis, Visualization, Writing—review and editing; Lihua Wang, David L Hunt, Dimitri Tränkner, Jayaram Chandrashekar, Serge Picard, Andrew L Lemire, Resources, Investigation; Nelson Spruston, Supervision, Funding acquisition, Project administration, Writing—review and editing; Adam W Hantman, Conceptualization, Resources, Data curation, Supervision, Funding acquisition, Project administration, Writing—review and editing; Sacha B Nelson, Conceptualization, Resources, Data curation, Formal analysis, Supervision, Funding acquisition, Writing—original draft, Project administration, Writing—review and editing

## Author ORCIDs
Ken Sugino http://orcid.org/0000-0002-5795-0635
Erin Clark https://orcid.org/0000-0002-4013-325X
Bryan M Hooks http://orcid.org/0000-0003-0135-4284
Jayaram Chandrashekar http://orcid.org/0000-0001-6412-0114
Nelson Spruston http://orcid.org/0000-0003-3118-1636
Sacha B Nelson http://orcid.org/0000-0002-0108-8599

## Ethics

Animal experimentation: All experiments were conducted in accordance with the requirements of the Institutional Animal Care and Use Committees at Janelia Research Campus (protocol# not available) and Brandeis University (protocol#17001).

## Decision letter and Author response

Decision letter https://doi.org/10.7554/eLife.38619.047
Author response https://doi.org/10.7554/eLife.38619.048

# Additional files

## Supplementary files

• Supplementary file 1. Table listing information for mouse lines. Information (columns) includes regions profiled, source of the mouse line, repository ID and URL, whether atlas is available via the Janelia viewer, URL for other atlases, and relevant references.
DOI: https://doi.org/10.7554/eLife.38619.029

• Supplementary file 2. Table for sample information. Included fields are, 1. sample_id: Sample ID; 2. sample_name: Sample Name; 3. group: Sample Group ID; 4. group_label: Label for Group; 5. sample_label: Label for Sample; 6. seqlane: Sequencing Lane ID; 7. mouseline: Mouse Line ID; 8. sample_code: Type of sample, cs.n: cell-type-specific neuronal sample; cs.o: cell-type-specific nonneuronal sample; ti.b: tissue sample from brain; ti.o: sample from non-brain tissue; cs.p: cell-type-specific progenitor sample; 9. region: Anatomical Region (large structure); 10. transmitter: Transmitter; 11. allenregion: Region using Allen Reference Atlas notation; 12. num_cells: Number of cells used in the sample; 13. age_(day): Postnatal age (in days) of the mouse; 14. sex: Sex of the mouse; 15. weight_(g): Weight (g) of the mouse; 16. ercc($10^-$5 dilution ul): Amount of added ERCC in ul. ($10^{-5}$ diluted); 17. ercc_mix: Which ERCC mix is used; 18. adaptor: Which Illumina (Solexa) sequencing adaptor is used; 19. total_reads: Total number of sequencing reads; 20. total_wo_ERCC: Total number of sequencing reads without reads mapping to ERCC; 21. read_length: Sequencing read length; 22. ercc%: Percentage of ERCC reads; 23. ribosomal_etc%: Percentage of reads mapping to ribosomal or other abundant sequences (phiX, polyC, polyA); 24. unmapped_reads%: Percentage of reads not mapped to mm10 genome; 25. unique_reads%: Percentage of reads uniquely mapped; 26. nonunique_reads%: Percentage of non-uniquely mapped reads; 27. short_insert%: Percentage of short (¡30 bp) reads; 28. mapped_reads: Number of mapped reads; 29. comments: Comments;
DOI: https://doi.org/10.7554/eLife.38619.030

• Supplementary file 3. Table listing public tissue samples used in analyses.
DOI: https://doi.org/10.7554/eLife.38619.031

• Transparent reporting form
DOI: https://doi.org/10.7554/eLife.38619.032

## Data availability

Sequencing data have been deposited in NCBI GEO under accession number GSE79238.

The following dataset was generated:

| Author(s) | Year | Dataset title | Dataset URL | Database and Identifier |
|---|---|---|---|---|
| Sugino K, Wang L, | 2018 | Transcriptional Basis of Neuronal | https://www.ncbi.nlm. | NCBI Gene |

| Author(s) | | Dataset title | Dataset URL | Database and Identifier |
|---|---|---|---|---|
| Shima Y, Hunt D, Lemire A, Hara E, Hooks M, Tränkner D, Chandrashekar J, Hantman A, Nelson S | | Diversity in the Mammalian Brain | nih.gov/geo/query/acc. cgi?acc=GSE79238 | Expression Omnibus, GSE79238 |

The following previously published datasets were used:

| Author(s) | Year | Dataset title | Dataset URL | Database and Identifier |
|---|---|---|---|---|
| Tasic B, Menon V, Nguyen TN, Kim TK, Yao Z, Gray LT, Hawrylycz M, Koch C, Zeng H | 2016 | Adult mouse cortical cell taxonomy by single cell transcriptomics | https://www.ncbi.nlm. nih.gov/geo/query/acc. cgi?acc=GSE71585 | NCBI Gene Expression Omnibus, GSE71585 |
| Zeisel A, Muñoz Manchado AB, Lönnerberg P, Linnarsson S | 2015 | Single-cell RNA-seq of mouse cerebral cortex | https://www.ncbi.nlm. nih.gov/geo/query/acc. cgi?acc=GSE60361 | NCBI Gene Expression Omnibus, GSE60361 |
| Zeisel A | 2018 | Molecular architecture of the mouse nervous system | https://storage.google-apis.com/linnarsson-lab-loom/l5_all.loom | Mouse brain atlas , l5_all.loom |
| Saunders A, Macosko E, Wysoker A, Goldman M | 2018 | A Single-Cell Atlas of Cell Types States, and Other Transcriptional Patterns from Nine Regions of the Adult Mouse Brain | https://storage.google-apis.com/dropviz-down-loads/static/metacells. BrainCellAtlas_Saun-ders_version_2018.04.01. csv | DropZiz , metacells. BrainCellAtlas_ Saunders_version_20 18.04.01.csv |
| Tasic B | 2018 | Cell Diversity in the Mouse Cortex | http://celltypes.brain-map.org/api/v2/well_known_file_download/ 694413985 | Allen Brain Map , 694413985 |

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

# Appendix 1

DOI: https://doi.org/10.7554/eLife.38619.033

## Relationship between DEF and Gini-Simpson index or MI

Here we explore in more detail the relationship between DEF (differentially expressed fraction of populations) and Gini-Simpson index (GSI) or MI (mutual information). DEF of a gene is equivalent to the Gini-Simpson index calculated using distinguishable levels of expression of the gene and it is also closely related to mutual information between (discretized) expression levels and cell population labels.

Assume there are $N_e$ distinguishable expression levels of a gene and there are $n_i$ cell population groups in level $i$. Then, the Gini-Simpson index (GSI) is:

$$GSI = 1 - \sum_{i=1}^{N_e} p_i^2 \tag{1}$$

$$= 1 - \sum_{i=1}^{N_e} \frac{n_i(n_i-1)}{N(N-1)} \tag{2}$$

Where $p_i$ is the probability of randomly selected element being in expression level $i$ and $N = \sum_{i=1}^{N_e} n_i$ is the total number of groups. The second equation holds since $p_i^2 = n_i(n_i-1)/N(N-1)$ for sampling without replacement.

Since $n_i(n_i-1)/N(N-1) = (n_i(n_i-1)/2)/(N(N-1)/2)$, this term is the fraction of pairs in level $i$. So the sum of these are the total fraction of indistinguishable pairs and one minus this sum equals the fraction of distinguishable pairs, which is DEF. Thus, DEF is equivalent to the Gini-Simpson index calculated using distinguishable levels of expression.

To calculate mutual information between expression levels and cell populations, we discretize expression levels into $N_e$ levels. Let $N_s$ be the number of samples. Let $n_{ij}$ be counts in the contingency table where $i = 1, ..., N_e$ and $j = 1, ..., N_s$. Then the joint probability distribution and the marginal probability distribution can be written as:

$$p(i,j) = \frac{n_{ij}}{N_s} \tag{3}$$

$$p(i) = \frac{\sum_j n_{ij}}{N_s} = \frac{n_i}{N_s} \tag{4}$$

$$p(j) = \frac{\sum_i n_{ij}}{N_s} = \frac{n_j}{N_s} \tag{5}$$

Where $n_i = \sum_j n_{ij}$ and $n_j = \sum_i n_{ij}$ is the number of samples in level $i$ and $n_j$ is the number of replicates in cell type $j$. The mutual information between expression level (E) and samples (S) is:

$$I(E;S) = \sum_{i,j} p(i,j) \log \frac{p(i,j)}{p(i)p(j)} \tag{6}$$

$$= \sum_{i,j} p(i,j) \log \frac{p(i,j)}{p(j)} - \sum_{i,j} p(i,j) \log p(i) \tag{7}$$

$$= \sum_{i,j} p(j)p(i|j) \log p(i|j) - \sum_{i,j} p(i,j) \log p(i) \tag{8}$$

$$= \sum_{j} p(j) \sum_{i} p(i|j) \log p(i|j) - \sum_{i} \log p(i) \sum_{j} p(i,j) \tag{9}$$

$$= -\sum_{j} p(j)H(E|S=j) - \sum_{i} p(i) \log p(i) \tag{10}$$

$$= -H(E|S) + H(E) \tag{11}$$

$H(E|S=j)$ is the entropy of expression levels in cell population j, which represents the expression noise in cell population j, and $H(E|S)$ is the average of these across all cell populations. When there are no replicates, $H(E|S)$ is zero. When there are replicates, $H(E|S=j)$ represents how noisy the expression is. This may depend on expression level, and $H(E|S)$, the average of $H(E|S=j)$ may depend on expression prevalence (i.e., how widely the gene is expressed), but in any case, the first term $-H(E|S)$ represents reduction of the mutual information by noise.

The second term $H(E)$ is the entropy of the marginal distribution $p(i)$ and represents the main information content about cell groups encoded in expression levels. This can be rewritten using counts in the contingency table as:

$$H(E) = -\sum_{i} p(i) \log p(i) \tag{12}$$

$$= -\sum_{i} \frac{n_i}{N_s} \log \frac{n_i}{N_s} \tag{13}$$

$$= -\sum_{i} \frac{n_i}{N_s} \log n_i + \sum_{i} \frac{n_i}{N_s} \log N_s \tag{14}$$

$$= -\frac{1}{N_s} \sum_{i} n_i \log n_i + \log N_s \tag{15}$$

Thus, it is maximized when all $n_i$'s are 0 or 1, which corresponds to the case in which one expression level corresponds to one cell population, making all cell populations distinguishable by the expression levels. This is true when the number of discretization levels exceeds the number of samples. When the number of discretization levels ($N_e$) is less than the number of samples ($N_s$), $H(E)$ takes the maximum value of $\log N_e$ when all the samples are distributed equally across each bin.

To explore the relationship between $H(E)$ and DEF, the $\log n_i$ in the first term is replaced (approximated) by $(n_i - 1)$ (first two terms in the Taylor expansion of $\log n_i$ around $n_i = 1$.):

$$H(E) \sim -\frac{1}{N_s} \sum_{i} n_i(n_i - 1) + \log N_s \tag{16}$$

$$= -\frac{2}{N_s} \sum_{i} n_i(n_i - 1)/2 + \log N_s \tag{17}$$

$$= \frac{2}{N_s} \left\{ N_s(N_s - 1)/2 - \sum_{i} n_i(n_i - 1)/2 \right\} - (N_s - 1) + \log N_s \tag{18}$$

$$= (N_s - 1)DEF - (N_s - 1) + \log N_s \tag{19}$$

Since $n_i$ is the number of samples in one expression level, $n_i(n_i - 1)/2$ is the number of indistinguishable pairs in that expression level when there are no replicates. The term within the curly bracket is then the number of distinguishable pairs, leading to *Equation (19)*.

More formally, since both $h(p) = \sum n_i \log n_i$ and $d(p) = \sum n_i(n_i - 1) = \sum n_i^2 - N_s$ are Schur-convex functions* on partitions of $N_s$, $p = (n_1, n_2, ..., n_k)$, when partition $p_1$ majorizes $p_2$ then, $h(p_1) \geq h(p_2)$ and $d(p_1) \geq d(p_2)$. When the partition length is 2, that is, when expression levels

are discretized into only two levels, corresponding to ON and OFF, then, all of the partitions can be ordered with respect to majorization, therefore, $h(p)$ and $d(p)$ are order-preserved transformations of each other (*Figure 3—figure supplement 1C* left). When the partition length is greater than 2, this relationship is not satisfied. However, they are still highly correlated to each other (*Figure 3—figure supplement 1C* right).

When DEF is calculated from global discretization (as in the above case), the maximum number of pairs distinguishable occurs when all samples are equally distributed across bins and the number of distinguishable pairs is $\left(\frac{N_s}{N_e}\right)^2 N_e(N_e-1)/2$. Therefore,

$$max(DEF) = \left(\frac{N_s}{N_e}\right)^2 \frac{N_e(N_e-1)/2}{N_s(N_s-1)/2} \tag{20}$$

$$= \left(1 - \frac{1}{N_e}\right) / \left(1 - \frac{1}{N_s}\right) \tag{21}$$

$$\sim 1 - \frac{1}{N_e} \quad (when \quad N_s \gg 1) \tag{22}$$

As stated above, this is also when the entropy $H(E)$ takes the maximum value of $\log_2 N_e$ in the unit of bits. (*Figure 3—figure supplement 1C*)

\* A Schur-convex function is a function $f : \mathbb{R}^k \rightarrow \mathbb{R}$ which satisfies $f(x) \geq f(y)$ for all $x, y$ where $x$ majorizes $y$. For $x = (x_1, x_2, ..., x_k) \in \mathbb{R}^k$ where $(x_1 \geq x_2 \geq ... \geq x_k)$ and $y = (y_1, y_2, ..., y_k) \in \mathbb{R}^k$ where $(y_1 \geq y_2 \geq ... \geq y_k)$, $x$ majorizes $y$ when $\sum_{i=1}^{k} x_i = \sum_{i=1}^{k} y_i$ and $\sum_{i=1}^{j} x_i \geq \sum_{i=1}^{j} y_i$ for all $j = 1, ..., k$. When $x$ majorizes $y$, it follows $x_i \geq y_i$ for all $i$, so it is easy to see $h(x) \geq h(y)$ and $d(x) \geq d(y)$.

