## [Decision Letter]

Thank you for submitting your article "The Transcriptional Logic of Mammalian Neuronal Diversity" for consideration by *eLife*. Your article has been reviewed by three peer reviewers, and the evaluation has been overseen by a Reviewing Editor and a Senior Editor. The following individual involved in review of your submission has agreed to reveal his identity: Nenad Sestan (Reviewer #2).

The reviewers have discussed the reviews with one another and the Reviewing Editor has drafted this decision to help you prepare a revised submission.

The reviewers described the broad survey of gene expression in nearly 200 mouse neuronal subpopulations as being a powerful and useful resource for the neuroscience community. A major strength was the use of transgenic labelled neuronal subpopulations with some anatomical information. Nevertheless, this Editor and some of the reviewers were not convinced by the analysis of "long genes" that gene length is an appropriate explanatory metric of regulatory complexity, particularly given that the density of regulatory elements across mammalian genomes is known to be relatively uniform. We consider the authors' hypothesis not to have been proved. Consequently, either this analysis needs to be removed or replaced or further substantiated. Additionally, we request that the comparison with single cell data is updated, and its presentation revised. Finally, it will be important to report findings using conventional statistical metrics rather than those currently used, unless the authors can justify that these are superior.

The following are the revisions that are required:

1) Currently only gene bodies, and not intergenic sequences, are considered in the analysis. Unless this subjective choice is further justified the authors will need to consider (e.g. for the ATAC-Seq analysis) all intergenic and intragenic regulatory elements. This could be done by considering a gene territory in its vicinity and ATAC-Seq peaks that connect to the gene via HiC peaks, for example. If a regulatory complexity metric is best explained by a gene's full (rather than intragenic) regulatory landscape then the associated speculation in the manuscript needs to be removed. Additionally, reviewers were not convinced: that the frequency of insertion mutations is uniform in the genome, as assumed; that sequence-similar, more cell/tissue-specific, paralogs could modulate this frequency; and, that there are specific population genetic studies that could test the authors' hypothesis.

2) The authors' attempt to reinvent the wheel, statistically, was considered unnecessary. Reporting results using conventional statistics should be sufficient. Attempts to develop novel test statistics should be removed unless with compelling justification. This is easily done since one of the statistics is close to "fraction of comparisons DE" and the other is close to a fold change. The NNLS method is itself not validated and also not essential for the analysis.

3) An important analysis is the comparison with single cell RNA-seq datasets (Figure 2). The problem with the current analysis is that the Ziesel, 2015 and Tasic, 2016 studies are already somewhat out of date, because they are pilot studies to the most recent Tasic et al., 2018 paper in bioRxiv which is already accepted. We understand that the dataset should be available soon, and thus that the timeline should be compatible with this revision. If the data set unexpectedly is not available please do let us know. Another high resolution dataset is from Paul et al., 2017, whose data should also be compared.

4) The use of "cell type" is misleading, even with "operational cell types" defined in the Materials and methods section. Even with known anatomic locations, it is quite likely that the labelled population comprises multiple "atomic types". "Subpopulation" is a more appropriate description and is no less significant. In this discussion, the authors must discuss the extent to which their analysis might be confounded by interregional or interindividual variation.

5) Figures. These could be further streamlined and shuffled to help the reader more. Figure 2 jumps straight into what may seem like an esoteric debate to those not currently diving into or weary of single cell RNA sequencing. A schematic of single cell vs. pooled cell analysis that illustrates shallow and deep RNA capture plus questions of cell purity might help introduce these heatmaps. Also, the Figure 2B panels could be placed in the supplement, and in Figure 2A, it might help to highlight examples of possible low purity cell types to highlight the overall very high purity of the data. Figure 3 – DI and SC will not be intuitive to all readers, even after the schematic in 2A. As a bridge, it might help to label individual genes with high SC and high DI from the scatterplot in B and show the expression of these genes across cell types. Figure 4A dives into this a level deeper by focusing on OFF noise, but Lhx1 and Calb2 could be labeled as examples in 3. Figure 5B: label how many long and short genes were considered.

6) The authors should supplement the use of PANTHER gene families to avoid, to the extent possible, biases in the specific datasets included in this database; it is possible that some gene families, including synaptic and signaling genes or homeobox transcription factors, are over-represented in this dataset. In Figure 4D, the PANTHER gene category "receptor" seems overly broad. What types of receptors? Certainly not all receptor genes are long – olfactory receptors are very short. Also, signaling proteins are on this list but are not discussed in the text along with ion channels and cell adhesion molecules. Why this selective avoidance?

[Editors' note: further revisions were requested prior to acceptance, as described below.]

Thank you for sending your article entitled "Mapping the transcriptional diversity of genetically and anatomically defined cell populations in the mouse brain" for peer review at *eLife*. Your article is being overseen by Chris Ponting, as Reviewing Editor, and Catherine Dulac as the Senior Editor.

Given the issue regarding "long genes" that we describe below, the editors and reviewers invite you to respond within the next two weeks with an action plan and timetable for the completion of additional work. We plan to discuss your response and then issue a binding recommendation.

You have placed considerable emphasis in your two versions of this manuscript on correlations with "long genes". With respect to these aspects (in many places, e.g. subsection “Long genes shape neuronal diversity”, fourth paragraph) it is our view that this argument cannot be retained unless you can satisfactorily refute the results of Raman et al., 2018. These indicate that the length dependencies that you, and others, have seen are likely a PCR artefact.

"Long genes shape neuronal diversity". This returns to our previous issue whether "long gene" expression is correlated with neuronal diversity or whether it causes neuronal diversity. The word "shape" implies causality, yet without evidence: their expression could be a consequence of diverse neuronal cell populations. You also have not shown whether these mRNA expression trends persist with proteins and this will need to be stated explicitly. "rich in transposons and other retroelements". This implies that these introns contain active elements whereas they contain the inactive debris of retrotransposons. The last paragraph of the Discussion is not warranted and should be excluded because it implies (without evidence) that long genes arise because of "exaptations".

---

## [Author Response]

The reviewers described the broad survey of gene expression in nearly 200 mouse neuronal subpopulations as being a powerful and useful resource for the neuroscience community. A major strength was the use of transgenic labelled neuronal subpopulations with some anatomical information. Nevertheless, this Editor and some of the reviewers were not convinced by the analysis of "long genes" that gene length is an appropriate explanatory metric of regulatory complexity, particularly given that the density of regulatory elements across mammalian genomes is known to be relatively uniform. We consider the authors' hypothesis not to have been proved. Consequently, either this analysis needs to be removed or replaced or further substantiated. Additionally, we request that the comparison with single cell data is updated, and its presentation revised. Finally, it will be important to report findings using conventional statistical metrics rather than those currently used, unless the authors can justify that these are superior.

In keeping with your suggestions, we changed the title and rewrote much of the manuscript to remove any attempt to provide evidence that regulatory complexity was responsible for the observed long gene bias. As requested we updated the single cell analyses and recast our statistical analyses in terms of the traditional metrics of fold-change and differential expression. In addition, we added a new analysis of alternative splicing. We have made very extensive changes in the manuscript and have attempted to respond to each point raised in the reviews as outlined below.

The following are the revisions that are required:1) Currently only gene bodies, and not intergenic sequences, are considered in the analysis. Unless this subjective choice is further justified the authors will need to consider (e.g. for the ATAC-Seq analysis) all intergenic and intragenic regulatory elements. This could be done by considering a gene territory in its vicinity and ATAC-Seq peaks that connect to the gene via HiC peaks, for example. If a regulatory complexity metric is best explained by a gene's full (rather than intragenic) regulatory landscape then the associated speculation in the manuscript needs to be removed. Additionally, reviewers were not convinced: that the frequency of insertion mutations is uniform in the genome, as assumed; that sequence-similar, more cell/tissue-specific, paralogs could modulate this frequency; and, that there are specific population genetic studies that could test the authors' hypothesis.

As requested, we removed the data on intronic peaks and the analyses of the differential presence of these peaks across cell types as a metric of regulatory complexity. We removed mention of any population genetic test of hypotheses about increased regulatory complexity.

2) The authors' attempt to reinvent the wheel, statistically, was considered unnecessary. Reporting results using conventional statistics should be sufficient. Attempts to develop novel test statistics should be removed unless with compelling justification. This is easily done since one of the statistics is close to "fraction of comparisons DE" and the other is close to a fold change.

As suggested, we recast our metrics to make it clearer that they are simply versions of Differential Expression and fold-change for simultaneous comparisons across many populations. The index previously called Differentiation Index (DI) is now called Differentially Expressed Fraction (DEF) and the previous Signal Contrast (SC) is now Fold-Change Ratio (FCR). We hope that the text, arrangement of the figures and the new terms now make clearer how these are simple extensions of the standard statistics to the situation of making succinct comparisons across many populations.

The NNLS method is itself not validated and also not essential for the analysis.

NNLS is a standard method of matrix decomposition. Here we provide extensive validation of its use for decomposition of mixed expression profiles. These include:

- Cross validation (testing the ability of half of each dataset to decompose the other half): Figure 2—figure supplement 1;

- A test of the ability to retrieve merged profiles: Figure 2—figure supplement 2;

- Comparison to another standard algorithm for decomposition, a random forest classifier: Figure 2—figure supplement 6.

This is the core method we used for comparison with single cell datasets, which is stated to be “an important analysis” hence it is essential for our analysis. These analyses have (as requested below) been fully updated to include more recent datasets. The conclusion, supported by these analyses is that there is a similar level of heterogeneity apparent from the decomposition of NeuroSeq sorted data by single cell profiles as is found in the decomposition of one set of single cell profiles by another. This is a key result of the paper.

3) An important analysis is the comparison with single cell RNA-seq datasets (Figure 2). The problem with the current analysis is that the Ziesel, 2015 and Tasic, 2016 studies are already somewhat out of date, because they are pilot studies to the most recent Tasic et al., 2018 paper in bioRxiv which is already accepted. We understand that the dataset should be available soon, and thus that the timeline should be compatible with this revision. If the data set unexpectedly is not available please do let us know. Another high resolution dataset is from Paul et al., 2017, whose data should also be compared.

We updated the analyses to include Tasic et al., 2018, Zeisel et al., 2018 and Paul et al., 2017. These are included in Figure 2 and in Figure 2—figure supplements 1, 3-7. In addition, we added comparisons to these datasets, as well as to Saunders et al., 2018 to Figure 5—figure supplement 3 (Off noise) and Figure 7—figure supplements 1 and 2 (length bias).

4) The use of "cell type" is misleading, even with "operational cell types" defined in the Materials and methods section. Even with known anatomic locations, it is quite likely that the labelled population comprises multiple "atomic types". "Subpopulation" is a more appropriate description and is no less significant. In this discussion, the authors must discuss the extent to which their analysis might be confounded by interregional or interindividual variation.

As requested, we have refrained from referring to the samples profiled as originating from cell types and have instead referred to them as Genetically- and Anatomically-identified Cell Populations (GACPs) or simply as “populations”. While we find this terminology cumbersome and not representative of the common use of the term “cell type” in the literature, we defer to the reviewers on this point.

5) Figures. These could be further streamlined and shuffled to help the reader more. Figure 2 jumps straight into what may seem like an esoteric debate to those not currently diving into or weary of single cell RNA sequencing. A schematic of single cell vs. pooled cell analysis that illustrates shallow and deep RNA capture plus questions of cell purity might help introduce these heatmaps. Also, the Figure 2B panels could be placed in the supplement, and in Figure 2A, it might help to highlight examples of possible low purity cell types to highlight the overall very high purity of the data.

We thank the reviewer for the suggestion to include a schematic in Figure 2. We think this helps us make the important point that apparent heterogeneity can arise both from pooling, as in our study, and from differences in the way clustering is applied in single cell studies. We think that this idea will be useful to readers of this paper and more generally to the literature on cell type-specific gene expression. As requested, we moved the Figure 2B panels to the supplement (now Figure 2—figure supplements 3, 4, and 5, which includes the Paul et al., 2017 data). Finally, we highlight examples of possible low purity cell types in the text of the Results section in the context of describing the cross validation (Figure 2—figure supplement 1).

Figure 3 – DI and SC will not be intuitive to all readers, even after the schematic in 2A. As a bridge, it might help to label individual genes with high SC and high DI from the scatterplot in B and show the expression of these genes across cell types.

To increase the clarity of this section, we renamed DI and SC to indicate their direct relationship to standard measures Differential Expression and Fold Change, and separated the explanation of these relationships (Figure 3) from the application to the data (Figure 4). As requested, we labeled specific example genes in Figure 4A and show their patterns of expression across populations in Figures 4B and C.

Figure 4A dives into this a level deeper by focusing on OFF noise, but Lhx1 and Calb2 could be labeled as examples in 3. Figure 5B: label how many long and short genes were considered.

We found other clearer examples to label in what was previously Figure 3 (now 4, see immediately above) but maintain the previously used examples in the now Figure 5. The numbers of long and short genes in the now Figure 6 are listed in the figure legend.

6) The authors should supplement the use of PANTHER gene families to avoid, to the extent possible, biases in the specific datasets included in this database; it is possible that some gene families, including synaptic and signaling genes or homeobox transcription factors, are over-represented in this dataset. In Figure 4D, the PANTHER gene category "receptor" seems overly broad. What types of receptors? Certainly not all receptor genes are long – olfactory receptors are very short. Also, signaling proteins are on this list but are not discussed in the text along with ion channels and cell adhesion molecules. Why this selective avoidance?

To avoid the possibility of implicit overrepresentation in the categories used for over-representation analyses, we used three different sets of categories. HUGO, PANTHER and Gene Ontology (specifically, GOM, the Gene Ontology Molecular Function categories). We replaced the use of PANTHER in the main figures with HUGO gene groups in most cases, and supplemented the analysis with PANTHER and GOM in the supplementary figures (Figure 4—figure supplement 1, Figure 5—figure supplement 1, Figure 5—figure supplement 3). In each case, the categories listed are those that exceeded the stated significance test, regardless of how broad or narrow these categories are. The use of HUGO to some degree mitigates some of the very broad categories found with PANTHER and Gene Ontology, because of the deeper hierarchies present in these systems of annotation. Nevertheless, some HUGO gene groups are also quite broad.

We note, for example, the Paul et al., 2017 paper uses HUGO but that Figure 2C of this paper contains categories that are equally broad: specifically: “Receptors” and “Signaling.” With respect to our own analyses, Figure 4E lists the categories GABA receptors, GPCRs, Glutamate receptors, GABA A receptors, Amine receptors etc. We now explicitly mention signaling genes in the Results section concerning Figure 4, in the legend of Figure 5 and in the Discussion, where we also cite the related finding by Paul et al., 2017.

[Editors' note: further revisions were requested prior to acceptance, as described below.]You have placed considerable emphasis in your two versions of this manuscript on correlations with "long genes". With respect to these aspects (in many places, e.g. subsection “Long genes shape neuronal diversity”, fourth paragraph) it is our view that this argument cannot be retained unless you can satisfactorily refute the results of Raman et al., 2018. These indicate that the length dependencies that you, and others, have seen are likely a PCR artefact.

We have now performed the Raman et al., 2018, test and include this as Figure 7—figure supplement 2. As can be seen, our results are highly significant even using this flawed test. We also refer to this result in the main text (subsection “Long genes contribute disproportionately to neuronal diversity”, second paragraph) and in the Discussion (subsection “Long genes and neuronal diversity”, first paragraph). As noted in our previous reply, we are preparing a more detailed rebuttal of the Raman et al. paper and so only include here our analysis of the fact that this cannot account for our observations of length biases.

"Long genes shape neuronal diversity". This returns to our previous issue whether "long gene" expression is correlated with neuronal diversity or whether it causes neuronal diversity. The word "shape" implies causality, yet without evidence: their expression could be a consequence of diverse neuronal cell populations.

We removed the word shape, changing the section title to “Long genes and neuronal diversity.” We also added the word “transcriptional” to clarify the following sentence: “Our study suggests that long neuronal effector genes contribute disproportionately to neuronal transcriptional diversity”.

You also have not shown whether these mRNA expression trends persist with proteins and this will need to be stated explicitly.

We added the following sentence: “However, we and others did not measure cell type-specific protein expression, and so cannot be sure that the long gene bias extends to the level of neuronal proteins.”

"rich in transposons and other retroelements". This implies that these introns contain active elements whereas they contain the inactive debris of retrotransposons.

We changed the sentence to: “These genes are long because of long introns that have accumulated sequences derived from transposons and other retroelements (Grishkevich et al., 2014).”

The last paragraph of the Discussion is not warranted and should be excluded because it implies (without evidence) that long genes arise because of "exaptations".

We have rewritten the final paragraph of the Discussion to present a more balanced account of this as only one of three possibilities.